



# A comprehensive porewater isotope model for simulating benthic nitrogen cycling: Description, application to lake sediments, and uncertainty analysis

Alessandra Mazzoli[1], Peter Reichert[2*], Claudia Frey[1], Cameron M. Callbeck[1], Tim J. Paulus[1], Jakob Zopfi[1], Moritz F. Lehmann[1]

[1]Department of Environmental Sciences, University of Basel, Basel, 4056, Switzerland
[2]Eawag, Swiss Federal Institute of Aquatic Science and Technology, Dübendorf, 8600, Switzerland
[*]Current status: retired from Eawag; email peter.reichert@emeriti.eawag.ch, see https://peterreichert.github.io for updated information

*Correspondence to*: Alessandra Mazzoli (alessandra.mazzoli@unibas.ch)



## Abstract

The combination of various nitrogen (N) transformation pathways (mineralization, nitrification, denitrification, DNRA, anammox) modulates the fixed-N availability in aquatic systems, with important environmental consequences. Several models have been developed to investigate specific processes and estimate their rates, especially in benthic habitats, known hotspots for N-transformation reactions. Constraints on the N cycle are often based on the isotopic composition of N species, which integrates signals from various reactions. However, a comprehensive benthic N-isotope model, encompassing all canonical pathways in a stepwise manner, and including nitrous oxide, was still lacking. Here, we introduce a new diagenetic N-isotope model to analyse benthic N processes and their N-isotopic signatures, validated using field data from the porewaters of the oligotrophic Lake Lucerne (Switzerland). As parameters in such a complex model cannot all uniquely be identified from sparse data alone, we employed Bayesian inference to integrate prior parameter knowledge with data-derived information. For parameters where marginal posterior distributions considerably deviated from prior expectations, we performed sensitivity analyses to assess the robustness of these findings. Alongside developing the model, we established a methodology for its effective application in scientific analysis. For Lake Lucerne, the model accurately replicated observed porewater N-isotope and concentration patterns. We identified aerobic mineralization, denitrification, and nitrification as dominant processes, whereas anammox and DNRA played a less important role in surface sediments. Among the estimated N isotope effects, the value for nitrate reduction during denitrification was unexpectedly low (2.8±1.1‰). We identified the spatial overlap of multiple reactions to be influential for this result.



## 1 Introduction

Nitrogen (N) is an essential element for all living organisms (Xu et al., 2022) and often limits primary production in aquatic
systems (Kessler et al., 2014). In order to meet the global demand for fixed N (nitrate, $NO_3^-$, and ammonium, $NH_4^+$),
industrial fixation of atmospheric dinitrogen ($N_2$) through the Haber-Bosch process now exceeds biological $N_2$ fixation, with
unforeseeable consequences regarding the ability of the environment to remove the excess fixed N, leaving the global N
cycle imbalanced (Kessler et al., 2014). High fixed-N in aquatic systems has detrimental environmental consequences (Denk
et al., 2017; Yuan et al., 2023), including eutrophication, ecosystem deterioration, and greenhouse gas emissions (e.g.,
nitrous oxide, $N_2O$). Thus, understanding the fate of fixed N in aquatic ecosystems and quantifying N fluxes are crucial for
global budget estimates (Pätsch and Kühn, 2008).
In aquatic systems, benthic habitats are important hotspots in the transformation of large amounts of fixed N (Dale et al.,
2019; Pätsch and Kühn, 2008; Xu et al., 2022), owing to sharp oxyclines and the co-occurrence of aerobic and anaerobic
processes. The active N cycle in these sediments is driven by the flux of organic matter (OM) from the photic zone along
with elevated concentrations of other electron donors (Ibánhez and Rocha, 2017; Wankel et al., 2015). Aerobic reactions,
such as nitrification (stepwise $NH_4^+$ oxidation to $NO_3^-$ via nitrite, $NO_2^-$, with $N_2O$ as by-product), are usually restricted to the
top few millimetres in OM-rich sediments (e.g., in small lakes) or extend several centimetres deep in OM-poor sediments
(e.g., in large oligotrophic lakes and the ocean) (Pätsch and Kühn, 2008; Wankel et al., 2015). The fate of $NO_3^-$, produced via
nitrification either locally in the sediments or in the water column, determines a system's capacity to function as an efficient
N sink (Wankel et al., 2015). Denitrification, the stepwise reduction of $NO_3^-$ to $N_2$ (via $NO_2^-$ and $N_2O$), has been identified as
a key pathway for anaerobic N removal. Additionally, anammox, the anaerobic oxidation of $NH_4^+$ to $N_2$ using $NO_2^-$, can
contribute to N loss (Ibánhez and Rocha, 2017; Kampschreur et al., 2012; Wankel et al., 2015), especially in oligotrophic
lake sediments (Crowe et al., 2017). In anammox, partial oxidation of $NO_2^-$ generates $NO_3^-$ as a by-product to provide
reducing equivalents for the fixation of inorganic carbon (C) (Brunner et al., 2013; Strous et al., 1999). Counteracting N
removal by anammox and denitrification, the dissimilatory $NO_3^-$ reduction to $NH_4^+$ (DNRA) contributes to N retention
(Denk et al., 2017; Ibánhez and Rocha, 2017; Rooze and Meile, 2016). The relative balance between these N-transforming
reactions is strongly influenced by environmental conditions, particularly the ratio of organic C to $NO_3^-$ and oxygen ($O_2$)
availability. For instance, DNRA may be predominant under high C:$NO_3^-$ ratios (Ibánhez and Rocha, 2017; Kraft et al.,
2011; Wang et al., 2020). Oxygen is a central regulator in this context: it controls the coupling of nitrification with
denitrification, anammox and DNRA, and modulates $N_2O$ production and consumption, with peak $N_2O$ yields typically
occurring at the oxic-anoxic interface (Ni et al., 2011). The spatial overlap of aerobic and anaerobic N cycling processes at
this transition zone in sediments often results in very low concentrations of metabolic intermediates (e.g., $N_2O$) in porewater,
complicating their measurements in natural benthic environments. This is particularly true for the analysis of natural-
abundance DIN isotopologues, which provide critical insights into N-cycling reactions and pathways. However, measuring
these isotopologues, especially low-concentration intermediates in porewater, is technically challenging, if not impossible at





all. To overcome these limitations, isotope modelling has become an essential tool for quantifying rapid N turnover at the
oxic-anoxic interface, and for evaluating environmental controls on N dynamics and isotope signatures across diverse
settings (Denk et al., 2017; Wankel et al., 2015).
Natural abundance stable isotope measurements can provide insights into the N cycle, and the fluxes within its pathways, as
microbial processes impart unique isotopic imprints on the involved N pools (Lehmann et al., 2003; Rooze and Meile, 2016;
Wankel et al., 2015). In most microbial processes, the isotopically lighter molecules are preferentially consumed, yielding
$^{15}$N-depleted products and $^{15}$N-enriched substrates (normal N-isotopic fractionation) (Kessler et al., 2014), with few
exceptions, such as $NO_2^-$ oxidation, which occurs with an inverse N isotope fractionation (Casciotti, 2009; Martin et al.,
2019). The isotopic composition of a given N pool is expressed in $\delta$-notation, $\delta^{15}$N (‰ vs. std) = [($R_{sample}/R_{std}$) – 1] x 1000,
where $R$ is the isotope ratio $^{15}$N/$^{14}$N, and the internationally recognized standard is atmospheric $N_2$ (Denk et al., 2017; Martin
et al., 2019). The extent of the isotopic fractionation for a reaction is quantified using the isotope effect, $\varepsilon$, defined as $\varepsilon$ (‰) =
[1 – ($^H k/^L k$)] x 1000, where $^H k$ and $^L k$ are the specific reaction rates for the isotopically heavy and light molecules,
respectively (Sigman and Fripiat, 2019). For instance, $\delta^{15}$N-$NO_2^-$ analysis can help differentiate reductive and oxidative
pathways of $NO_2^-$ consumption, as they are characterised by a normal and an inverse kinetic isotope effect, respectively
(Dale et al., 2019; Martin et al., 2019; Rooze and Meile, 2016). Despite considerable efforts to estimate isotope effects for
most N-transformation processes (Denk et al., 2017), isotope effects estimated in batch cultures often differ from in situ
measurements (Martin et al., 2019). To date, a comprehensive benthic isotope model that integrates multiple N-
transformation processes in a stepwise manner, and the expression of their isotope effects in the porewater of aquatic
sediments, validated with observational data, is still lacking (Denk et al., 2017).
Existing N-isotope models address specific aspects of the N cycle (Denk et al., 2017), such as denitrification (Kessler et al.,
2014; Lehmann et al., 2003; Wankel et al., 2015), $NO_2^-$ oxidation and reduction (Buchwald et al., 2018) or $N_2$O dynamics
(Ni et al., 2011; Wunderlin et al., 2012). As denitrification is the primary pathway for fixed-N loss in many aquatic systems,
models integrating dual $NO_3^-$ isotopes (Lehmann et al., 2003; Wankel et al., 2015) have been used for example, to constrain
its partitioning between water-column and benthic denitrification (Lehmann et al., 2005), as well as the contribution of
regenerated $NO_3^-$ supporting denitrification (Lehmann et al., 2004). Rooze and Meile (2016) combined isotope data with a
reaction-transport model to investigate the influence of hydrodynamics on fixed-N removal, highlighting enhanced coupling
of nitrification-$N_2$ production by benthic infauna. Buchwald et al. (2018) used dual $NO_3^-$ and $NO_2^-$ isotope analyses, and a
reaction-diffusion model to demonstrate the tight coupling of $NO_3^-$ reduction and $NO_2^-$ oxidation near oxic-anoxic interfaces,
emphasizing the central role of $NO_2^-$ in N recycling. In contrast, most $N_2$O modelling efforts (primarily concentration-based
models) to date have focused on engineered systems such as wastewater treatment, where they have been used to assess $N_2$O
production pathways under variable conditions, and to minimize its emissions (Ni et al., 2011; Wunderlin et al., 2012).
Challenges in measuring $N_2$O isotopologues in natural settings, especially in sediment porewaters, have limited the broader
application of $N_2$O isotopic approaches and led to the exclusion of $N_2$O from benthic N-isotope modelling efforts so far.
Nonetheless, given the key role of $N_2$O in the N cycle, and its sensitivity to redox conditions, there is a growing need for



modelling frameworks that integrate multi-species N-isotope dynamics, even in the absence of direct measurements of N-
cycle intermediate like $NO_2^-$ and $N_2O$ to more accurately capture the interconnected nature of N transformations in natural
systems.
With this study, we introduce a comprehensive 1-D diffusion-reaction model, encompassing all canonical N-transformation
processes and most DIN isotopologues, to assess the role of distinct environmental factors (e.g., sediment reactivity,
bioturbation) in shaping porewater N dynamics and the N isotopic signatures the different N transformations (and
combinations thereof) generate. Furthermore, by considering the stepwise nature of the N-cycling pathways, the model
quantifies and isotopically characterizes key intermediates (i.e., $N_2O$, $NO_2^-$), which serve as substrates for subsequent
reactions (Martin et al., 2019). Moreover, the model acts as a valuable research tool for analysing process couplings (e.g.,
DNRA-anammox interactions) (Dale et al., 2019; Hines et al., 2012), which are crucial for accurately estimating N removal
and recycling, and can influence the apparent isotope effects of $NO_3^-$ and $NO_2^-$. Incorporating $N_2O$ isotopologues as state
variables enables the model to resolve the relative importance of $N_2O$ producing mechanisms across small-scale benthic
oxic-anoxic interfaces, and to quantify their contribution to sedimentary $N_2O$ emissions.
The application of a comprehensive diagenetic N isotope model to measured porewater profiles of selected inorganic N
compounds often results in parameter identifiability issues. Specifically, similar fits to the observed data might be achieved
with comparable accuracy using different parameter sets, each yielding distinct transformation rates. To reduce the risk of
drawing erroneous conclusions from such identifiability problems, we employed the following modelling strategies:
• *Use of prior knowledge*
Prior knowledge informed both the development of the model structure and the selection of parameter values. The
model parameterization was adapted as deemed necessary to effectively integrate this prior knowledge. This
approach aims to produce a plausible representation of the mechanisms governing the data.
• *Consideration of uncertainty*
Uncertainty in model parameters was explicitly accounted for using epistemic probability distributions. Bayesian
inference (Bernardo and Smith, 1994; Gelman et al., 2013; Robert, 2007) was employed to combine prior
knowledge with information obtained from observational data. The resulting posterior distribution of the parameters
and calculated results provide a comprehensive uncertainty description, which is, however, still conditioned on prior
information about the model structure and parameters.
• *Sensitivity analysis*
To test the robustness of key results against modelling assumptions, we assessed their sensitivity to the choice of
prior probability distribution of the model parameters and to the inclusion of specific active processes within the
model.
Since the numerical implementation of Bayesian inference requires the computationally intensive Markov Chain Monte
Carlo (MCMC) sampling technique (Andrieu et al., 2003), an efficient model implementation is required. To meet this need,
we implemented the model in Julia (Bezanson et al., 2017) (https://julialang.org), a high-performance programming





language. This choice also enables the use of automatic differentiation, which supports advanced MCMC techniques like
Hamiltonian Monte Carlo (HMC) (Betancourt, 2017; Neal, 2011). The model was tested using field measurements from
oligotrophic Lake Lucerne. It is important to emphasize that this isotope model is designed as a research tool, rather than a
predictive instrument. Its primary purpose is to test hypotheses and assumptions related to the biogeochemical controls on N
isotope signatures in natural environments, and to assess the identifiability of process rates and N isotope effects from
observational data.

## 2 Model description

### 2.1 Model formulation

A one-dimensional diffusion-reaction model was developed to simulate the concentrations of inorganic N compounds ($NO_3^-$,
$NO_2^-$, $NH_4^+$, $N_2$, $N_2O$), distinguishing between $^{14}N$ and $^{15}N$ isotopes ($^{14}NO_3^-$, $^{15}NO_3^-$, $^{14}NO_2^-$, $^{15}NO_2^-$, $^{14}NH_4^+$, $^{15}NH_4^+$, $^{14}N_2$,
$^{14}N^{15}N$, $^{15}N_2$, $^{14}N_2O$, $^{14}N^{15}NO$, $^{15}N_2O$), as well as for $O_2$ and sulfate ($SO_4^{2-}$) concentrations. Their production and
consumption rates are described by incorporating key processes of the canonical N cycle: aerobic mineralization,
denitrification, nitrification, anammox, DNRA, mineralization by $SO_4^{2-}$ reduction, and anaerobic mineralization (other than
$SO_4^{2-}$-driven) (Fig. 1). All reactions (Table 1) are described using the general formula:

$$\text{rate} = k_{max} \cdot \text{limitation} \cdot \text{inhibition} \qquad (1)$$

where $k_{max}$ represents the maximum conversion rate under ideal conditions (in µM d$^{-1}$). The terms for limitation by substrate
X and inhibition by substance Y for the process i are defined following Michaelis-Menten kinetics (Martin et al., 2019):

$$\text{limitation} = \frac{[X]}{K_{X,i}+[X]} \qquad (2) \qquad\qquad \text{inhibition} = \frac{K_{Y,i}}{K_{Y,i}+[Y]} \qquad (3)$$

where $[X]$ and $[Y]$ are the concentrations (in µM) of substances X and inhibitor Y, respectively, while $K_{X,i}$ and $K_{Y,i}$ are their
respective half-saturation and inhibition constants (in µM) for process i, respectively. While the model supports exponential
equations for limitation and inhibition terms, Michaelis-Menten kinetics were chosen for this study, as they are more
commonly employed in N models (Rooze and Meile, 2016). The specific reaction rate equations are implemented taking into
account the concentrations of $^{14}N$, $^{15}N$, $^{14}N^{14}N$, $^{14}N^{15}N$, and $^{15}N^{15}N$ species separately for the limitation term. For $^{15}N$-
containing species, specific reaction rates are reduced by ($1-\varepsilon/1000$) relative to $^{14}N$-containing species, reflecting the isotope
effect associated with a given reaction (detailed descriptions of the model processes are provided in Appendix A: *Model*
*processes and stoichiometry*).
Molecular diffusion is modelled taking into account the reduced solute movement due to tortuosity (Burdige, 2007).
Additionally, bioturbation is included as a transport term enhancing diffusion, with its influence exponentially decreasing
with depth. Boundary conditions are set based on observed concentrations of N compounds, $O_2$, $SO_4^{2-}$ at the upper boundary,
and by zero fluxes at the lower boundary, except for $NH_4^+$, the flux of which was jointly estimated with the model



parameters. Total N, [14]N and [15]N concentrations, along with their fluxes, are used for model parameterization (see Appendix
B: *Reaction-diffusion model* for details).
The model is formulated as a dynamic model, but simulated to steady-state for comparison with observational data.
Concentrations of [14]N- and [15]N-containing compounds are converted to total concentrations and $\delta^{15}$N.

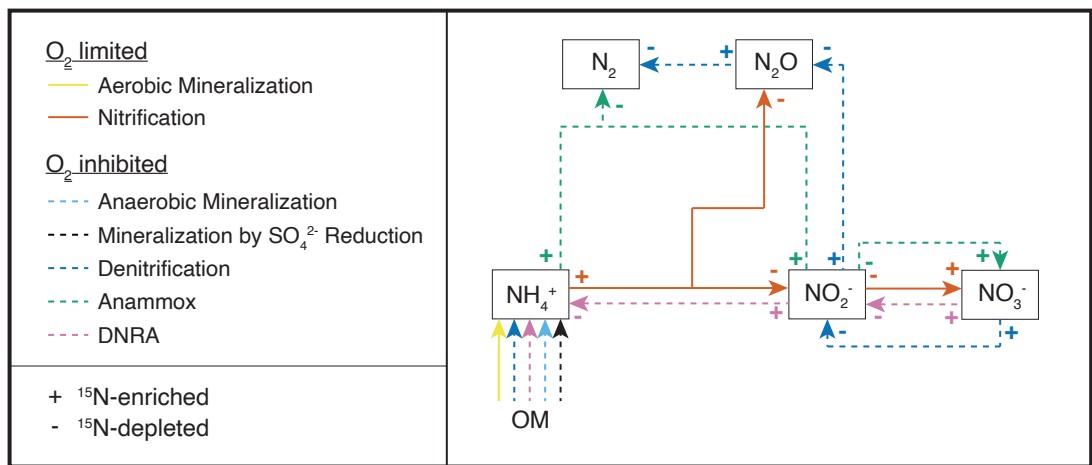


**Figure 1: Simplified scheme of the N-transformation reactions considered for the diagenetic isotope model described in this paper.**
**Continuous lines identify aerobic processes, while dashed lines indicate anaerobic processes. The state variables explicitly modelled**
**as substrates for the considered reactions are highlighted with outlined boxes; $O_2$ is modelled as a state variable and as a regulator**
**of aerobic and anaerobic processes; organic matter (OM) is not a state variable *per se* within the framework of this model, but acts**
**as a source of N for the remaining processes. The isotopic fractionation of each process is shown using + and – signs to represent**
**the $^{15}$N-enriching and $^{15}$N-depleting effects of the respective reactions.**

## 2.2 Description of modelled transformation processes

This section outlines the modelled processes for [14]N and [14]N[14]N compounds (Table 1). A comprehensive overview of the
transformation processes for all isotopologues, and stoichiometric relations is provided in Appendix A: *Model processes and*
*stoichiometry*.
Mineralization of OM, the sole external N source, is differentiated in the model according to the specific electron acceptor
involved: aerobic mineralization ($O_2$), denitrification and DNRA ($NO_3^-$), $SO_4^{2-}$ reduction, and anaerobic mineralization. The
latter encompasses all remaining redox species (i.e., other than $O_2$, $NO_3^-$, and $SO_4^{2-}$) below the nitracline (e.g., manganese,
iron, carbon dioxide).
Denitrification is modelled as a three-step process: (1) $NO_3^-$ to $NO_2^-$; (2) $NO_2^-$ to $N_2O$; and (3) $N_2O$ to $N_2$. The first step,
typically regarded as the rate-limiting step (Kampschreur et al., 2012), is the primary control on the overall expression of the
N isotope effect (Kessler et al., 2014; Rooze and Meile, 2016). To prevent unrealistic rates, subsequent steps are constrained
by setting $k_{Den2} = f_{Den2} \times k_{Den1}$ and $k_{Den3} = f_{Den3} \times k_{Den1}$, and specifying priors for $f_{Den2}$ and $f_{Den3}$. The $NO_3^-$ N isotope effect
during benthic denitrification is known to be suppressed in the overlying water due to diffusion limitation (Dale et al., 2022;
Kessler et al., 2014; Lehmann et al., 2003), though its expression at the porewater level remains less well constrained



(Wankel et al., 2015). Transiently accumulating intermediates, such as $N_2O$, that can escape to the overlying water and alter
benthic N fluxes (Rooze and Meile, 2016), are also considered. Lastly, to ensure mass balance, the model accounts for
clumped (doubly substituted; e.g., $^{15}N^{15}NO$ and $^{15}N^{15}N$) isotopocules, but does not distinguish between isotopomers (i.e.,
$^{14}N^{15}NO$ and $^{15}N^{14}NO$) due to lack of $N_2O$ isotope data needed for model validation. For the purpose of comparison with
previous N models, a simplified one-step denitrification pathway ($NO_3^-$ to $N_2$ with no release of $NO_2^-$ or $N_2O$ into the
environment) approach is also implemented in the model code.
Nitrification is modelled as a two-step process: (1a) $NH_4^+$ to $NO_2^-$; (1b) $NH_4^+$ to $N_2O$; (2) $NO_2^-$ to $NO_3^-$. As for
denitrification, the second step of nitrification is constrained to prevent unrealistic rates: $k_{Nit2} = f_{Nit2} \times k_{Nit1}$, with specifying a
prior for $f_{Nit2}$. $N_2O$ production yield during the first step is $O_2$-dependent, and is modelled accordingly:
$$f_{N2O\_Nit1} = \frac{b\,a}{[O_2]+a} \qquad (4)$$

where $b$ and $a$ are empirical parameters derived from (Ji et al., 2018). $N_2O$ production also occurs via nitrification-
denitrification, implicitly modelled by allowing reaction coupling via the intermediate $NO_2^-$. The expression of isotope
effects depends on substrate availability and reaction completion. For instance, incomplete nitrification has been shown to
result in isotopically heavy $NH_4^+$ efflux from the sediments (Dale et al., 2022; Lehmann et al., 2004; Rooze and Meile,
2016). However, similar phenomena for $N_2O$ and $NO_2^-$ remain poorly understood.
The limited understanding of porewater N isotope dynamics, especially for processes other than denitrification, hinges on the
scarcity of isotope data for crucial N species like $NH_4^+$ and $NO_2^-$ in natural settings (Martin et al., 2019; Wankel et al., 2015).
In the present model, we investigated the importance of these solutes, and how N-turnover processes like DNRA and
anammox shape the distribution of their N isotopes. DNRA is modelled as a two-step process: (1) $NO_3^-$ to $NO_2^-$; and (2)
$NO_2^-$ to $NH_4^+$. This approach separates the impact of $NO_2^-$ reduction on $NH_4^+$, and allows comparison of $NO_2^-$ isotopic
signatures induced by denitrification, DNRA, and anammox. Anammox is modelled to include $NO_3^-$ production via $NO_2^-$
oxidation (0.3 mol $NO_3^-$ produced per 1 mol $NH_4^+$ and 1.3 mol $NO_2^-$) (Martin et al., 2019), which imparts a strong inverse
isotope fractionation (Brunner et al., 2013; Magyar et al., 2021).
The relative importance of reductive $NO_3^-$ pathways is constrained by altering maximum conversion rates, $k$, as: $k_{DNRA1} =$
$f_{DNRA1,Den1} \times k_{Den1}$; $k_{DNRA2} = f_{DNRA2,Den2} \times k_{Den2}$; $k_{Anam} = f_{Anam,Den2} \times k_{Den2}$, where prior information on $f$ factors was obtained from
experimental rate measurements (see below). Altogether these reactions provide a comprehensive overview of N isotope
dynamics in porewater and enable the assessment of influential environmental conditions in shaping them.
**Table 1: Chemical equations and reaction rate formulations for $^{14}N$ and $^{14}N^{14}N$ compounds across all modelled processes. The**
**rates for $^{15}N$, $^{15}N^{14}N$, and $^{15}N^{15}N$ are formulated analogously by replacing the concentration of the isotopologue of interest as**
**needed. The turnover rates for $^{15}N$-containing species are scaled by a factor of $(1-\varepsilon/1000)$, as outlined in the text. The complete set**
**of equations including all isotopic compositions, and the process stoichiometry is provided in Appendix A: *Model processes and***
***stoichiometry*. Anaerobic mineralization encompasses OM degradation coupled to iron and manganese reduction, as well as**
**through methanogenesis.**

| Reaction | Equation | Reaction rate |
|---|---|---|
| | | |





---

**Aerobic mineralization**

$$C_{106}H_{263}O_{110}N_{16}P + 106O_2 \rightarrow 106HCO_3^- + 16NH_4^+ + HPO_4^{2-} + 92H^+$$

$$r_{MinOx} = k_{MinOx} \frac{[O_2]}{K_{O2,MinOx} + [O_2]}$$

**Anaerobic Mineralization**

$$C_{106}H_{263}O_{110}N_{16}P + 212MnO_2 + 120H_2O \rightarrow 106HCO_3^- + 16NH_4^+ + HPO_4^{2-} + 212Mn^{2+} + 332OH^-$$

$$C_{106}H_{263}O_{110}N_{16}P + 424FeOOH + 120H_2O \rightarrow 106HCO_3^- + 16NH_4^+ + HPO_4^{2-} + 424Fe^{2+} + 332OH^-$$

$$C_{106}H_{263}O_{110}N_{16}P \rightarrow 53CH_4^+ + 53HCO_3^- + 16NH_4^+ + HPO_4^{2-} + 53H_2O + 14H^+$$

$$r_{MinAnae} = k_{MinAnae} \frac{K_{NO3,MinAnae}}{K_{NO3,MinAnae} + [^{14}NO_3^-] + [^{15}NO_3^-]} \frac{K_{O2,MinAnae}}{K_{O2,MinAnae} + [O_2]}$$

**Sulfate Reduction coupled to Mineralization**

$$C_{106}H_{263}O_{110}N_{16}P + 53SO_4^{2-} + 15H^+ \rightarrow 106HCO_3^- + 16NH_4^+ + HPO_4^{2-} + 53H_2S$$

$$r_{MinSulfRed} = k_{MinSulfRed} \frac{K_{NO3,MinSulfRed}}{K_{NO3,MinSulfRed} + [^{14}NO_3^-] + [^{15}NO_3^-]} \frac{K_{O2,MinSulfRed}}{K_{O2,MinSulfRed} + [O_2]} \frac{[SO_4^{2-}]}{K_{SO4,MinSulfRed} + [SO_4^{2-}]}$$

**Nitrification**   [1a]   $NH_4^+ + 1.5O_2 \rightarrow NO_2^- + 2H^+ + H_2O$

$$r_{Nit1a} = k_{Nit1}(1 - f_{N2O,Nit1}) \frac{[^{14}NH_4^+]}{K_{NH4,Nit1} + [^{14}NH_4^+] + [^{15}NH_4^+]} \frac{[O_2]}{K_{O2,Nit1} + [O_2]}$$

[1b]   $NH_4^+ + O_2 \rightarrow 0.5N_2O + H^+ + 1.5H_2O$

$$r_{Nit1b} = k_{Nit1} f_{N2O,Nit1} \frac{[^{14}NH_4^+][^{14}NH_4^+]}{(K_{NH4,Nit1} + [^{14}NH_4^+] + [^{15}NH_4^+])^2} \frac{[O_2]}{K_{O2,Nit1} + [O_2]}$$

[2]   $NO_2^- + 0.5O_2 \rightarrow NO_3^-$

$$r_{Nit2} = k_{Nit2} \frac{[^{14}NO_2^-]}{K_{NO2,Nit2} + [^{14}NO_2^-] + [^{15}NO_2^-]} \frac{[O_2]}{K_{O2,Nit2} + [O_2]}$$

**Denitrification**   [1]   $5C_{106}H_{263}O_{110}N_{16}P + 424NO_3^- \rightarrow 212HCO_3^- + 32NH_4^+ + 2HPO_4^{2-} + 424NO_2^- + 184H^+ + 3C_{106}H_{263}O_{110}N_{16}P$

$$r_{Den1} = k_{Den1} \frac{[^{14}NO_3^-]}{K_{NO3,Den1} + [^{14}NO_3^-] + [^{15}NO_3^-]} \frac{K_{O2,Den1}}{K_{O2,Den1} + [O_2]}$$

[2]   $3C_{106}H_{263}O_{110}N_{16}P + 424NO_2^- + 240H^+ \rightarrow 212HCO_3^- + 32NH_4^+ + 2HPO_4^{2-} + 212N_2O + 212H_2O +$
$$C_{106}H_{263}O_{110}N_{16}P$$

$$r_{Den2} = k_{Den2} \frac{[^{14}NO_2^-]}{(K_{NO2,Den2} + [^{14}NO_2^-] + [^{15}NO_2^-])^2} \frac{K_{O2,Den2}}{K_{O2,Den2} + [O_2]}$$

[3]   $C_{106}H_{263}O_{110}N_{16}P + 212N_2O \rightarrow 106HCO_3^- + 16NH_4^+ + HPO_4^{2-} + 212N_2 + 92H^+$

$$r_{Den3} = k_{Den3} \frac{[^{1414}N_2O]}{K_{N2O,Den3} + [^{1414}N_2O] + [^{1415}N_2O] + [^{1515}N_2O]} \frac{K_{O2,Den3}}{K_{O2,Den3} + [O_2]}$$



DNRA     [1]    $C_{106}H_{263}O_{110}N_{16}P + 212NO_3^- \rightarrow 106HCO_3^- + 16NH_4^+ + HPO_4^{2-} + 212NO_2^- + 92H^+$

$$r_{DNRA1} = k_{DNRA1} \frac{[^{14}NO_3^-]}{K_{NO3,DNRA1} + [^{14}NO_3^-] + [^{15}NO_3^-]} \frac{K_{O2,DNRA1}}{K_{O2,DNRA1} + [O_2]}$$

[2]    $3C_{106}H_{263}O_{110}N_{16}P + 212NO_2^- + 212H_2O + 148H^+ \rightarrow 318HCO_3^- + 260NH_4^+ + 3HPO_4^{2-}$

$$r_{DNRA2} = k_{DNRA2} \frac{[^{14}NO_2^-]}{K_{NO2,DNRA2} + [^{14}NO_2^-] + [^{15}NO_2^-]} \frac{K_{O2,DNRA2}}{K_{O2,DNRA2} + [O_2]}$$

Anammox     $NH_4^+ + 1.3NO_2^- + 0.15CO_2 \rightarrow N_2 + 0.3NO_3^- + 0.15CH_2O + 1.85H_2O$

$$r_{Anam} = k_{Anam} \frac{[^{14}NH_4^+]}{K_{NH4,Anam} + [^{14}NH_4^+] + [^{15}NH_4^+]} \frac{[^{14}NO_2^-]}{K_{NO2,Anam} + [^{14}NO_2^-] + [^{15}NO_2^-]} \frac{K_{O2,Anam}}{K_{O2,Anam} + [O_2]}$$

## 2.3 Model assumptions

The model builds on the following considerations and assumptions:

i.  The inputs of sinking OM and associated advective transport relative to the sediment surface are not explicitly modelled, as the dissolved $O_2$ and N-compound profiles tend to reach quasi-steady state on short timescales (days to weeks). This simplification may not be valid for continental shelf sediments, where advection dominates solute movement due to high sediment permeability (Rooze and Meile, 2016). Therefore, in our model, porewater profiles are shaped primarily by molecular diffusion and bioturbation (the latter approximated as enhanced diffusion), along with reaction processes.

ii.  Hinging on assumption i., the rates of OM-degrading processes are assumed to be limited by the availability of oxidants and not of OM, as in Kessler et al. (2014).

iii.  Microorganisms involved in N-transformation pathways are not explicitly modelled, meaning that maximum conversion rates, $k$, represent a combination of bacterial maximum specific growth rates and abundance. These parameters likely vary significantly across systems, due to differences in OM loading.

iv.  N assimilation is not included, which is plausible if the turnover rates of the modelled processes are considerably higher than the N assimilation rates.

v.  Maximum specific conversion rates for all reactions are constant with depth, implying uniform bacterial abundance and activity across the sediment layer affected by any given process.

vi.  Limitation and inhibition kinetics are modelled using Michaelis-Menten functions, as they are commonly employed in N-cycle models (Rooze and Meile, 2016); exponential equations are provided within the code as an alternative approach, depending on user preference.

vii.  OM composition is approximated by the Redfield ratio (C:N:P = 106:16:1), used to estimate the fraction of $NH_4^+$ released during OM mineralization.



239 viii. Anaerobic mineralization includes all processes involving redox species below the nitracline (e.g., manganese, iron,

240  and carbon dioxide) with the exception of $SO_4^{2-}$ reduction, with no distinction in reaction rate for different oxidants.

241  Reduction of $SO_4^{2-}$ is modelled separately, as it can occur at faster rates than oxidation by iron(III), $Fe^{3+}$, and

242  manganese, $Mn^{3+}$, in some lacustrine systems (Steinsberger et al., 2020), and is the dominant anaerobic

243  mineralization process in marine settings.

244 ix. N isotope effects for all processes are kept constant across depth and substrate availability.

245 x. OM mineralization occurs with no N isotopic fractionation; that is, the released $NH_4^+$ has the same N isotopic

246  composition of OM, which is a model parameter considered for estimation.

247 xi. Diffusivities of isotopologues are considered identical, as their differences have been reported to be minimal

248  (Lehmann et al., 2007; Wankel et al., 2015).

249 xii. Bioturbation enhances diffusion equally for all modelled species.

250 xiii. The yield of $NO_3^-$ during anammox is fixed at 0.3 mol $NO_3^-$ per 1 mol $NH_4^+$, although reported values range from

251  0.26 to 0.32 (Brunner et al., 2013).

252 xiv. The $NO_3^-$ and $NO_2^-$ equilibrium during anammox has been previously reported to occur under environmental stress

253  conditions with a strong isotopic fractionation (up to -60.5‰) (Brunner et al., 2013). Since it leads to the production

254  of $^{15}N$-enriched $NO_3^-$, similarly to the kinetic isotopic fractionation during $NO_2^-$ oxidation to $NO_3^-$, variable values

255  of $\varepsilon_{Anam,side}$ (-15‰ to -45‰) can encompass both kinetic and equilibrium fractionation.

256 xv. $NH_4^+$ adsorption and desorption rates are assumed to be comparable, and to occur with negligible isotopic

257  fractionation, resulting in no net effect on the $NH_4^+$ pool concentration or isotopic composition.

258 The model incorporates deliberate simplifications to reduce complexity, while remaining adaptable to new data or insights;

259 however, it is acknowledged that these assumptions may significantly influence model outcomes and should be carefully

260 considered when interpreting results.

**2.4 Prior knowledge about model parameters**

262 Model parameter values were derived from an extensive literature review, and formulated as prior distributions, as detailed

263 and referenced in Appendix C: *Prior values for inference*. Positive parameters were parameterized as Lognormal priors,

264 while priors of positive or negative parameters were parameterized as Normal distributions. Mean values were derived from

265 the provided references, standard deviations were assigned either as absolute values or as percentages of the mean,

266 depending on the class of variables. For parameters that are lake-specific (see model assumption iii.) and expected to be well

267 identifiable from data, such as the maximum conversion rates of various processes (i.e., aerobic mineralization, the first step

268 of nitrification, the first step of denitrification, mineralization by $SO_4^{2-}$ reduction, anaerobic mineralization) and the $NH_4^+$

269 flux from deeper sediment layers, only limited prior knowledge is available, making the use of uniform priors preferable. As

270 their interpretability can be questionable, uniform priors were applied only to parameters expected to be well-identifiable,





ensuring that prior variations within the marginal posterior range would remain small, even with alternative broad priors.
This approach avoids specifying typical expected values, while maintaining robust identifiability. The maximum conversion
rates for anammox, DNRA, as well as the second step of nitrification and the second and third steps of denitrification
(Anam, DNRA1, DNRA2, Nit2, Den2 and Den3) were more challenging to identify from data, as the sensitivity of model
results to these parameters becomes very low when the concentration of the converted substance becomes small.
Additionally, prior specification for these rates was difficult, due to the expected variability among different lakes, similar to
other maximum conversion rate parameters. Therefore, their priors were formulated as ratios relative to the better-
constrained maximum conversion rate of the first nitrification (i.e., $k_{Nit1}$) or denitrification step (i.e., $k_{Den1}$). This approach
allowed for the characterization of the relative importance of each process without requiring absolute rate values. The joint
prior for all parameters was assumed to be an independent combination of their respective marginal prior distributions.
**2.5 Model-based analysis process**
To partially reduce structural uncertainty of the model and to account for parameter non-identifiability, Bayesian inference
was applied, considering all uncertain parameters listed in Appendix C: *Prior values for inference*. Some parameters were
excluded from this analysis, including molecular diffusion coefficients, compound concentrations at the sediment surface,
zero fluxes from deeper sediment layers (except for the $NH_4^+$ flux, which was inferred jointly with other parameters) and
bioturbation. These values are considerably less uncertain than the other model parameters, except for bioturbation, which
was addressed separately through a scenario analysis, following Bayesian inference under the Base scenario.
The posterior distribution (probability density) of the model parameters, $f_{post}$, is expressed as
$$f_{\text{post}}(\theta) = \frac{f_L(C|\theta)\, f_{\text{pri}}(\theta)}{\int f_L(C|\theta')\, f_{\text{pri}}(\theta')\, d\theta'} \qquad (5)$$

where $f_{pri}$ is the prior distribution (probability density) of the model parameters, $f_L(C|\theta)$ is the likelihood function of the
model, $C$ represents the observed compound concentrations, or $\delta^{15}N$ values, and $\theta$ denotes the model parameters. The
likelihood function $f_L(C|\theta)$ is defined as a multivariate, uncorrelated Normal distribution with constant variances (standard
deviation, $\sigma_\delta$) for $\delta^{15}N$ values, and variances increasing linearly with concentration, leading to a standard deviation $\sigma_C =$
$\sqrt{\sigma_{C,a}\, C + \sigma_{C,b}{}^2}$ for $O_2$, $SO_4^{2-}$, and N compound concentrations. This formulation incorporates the combined uncertainties in
model structure, sampling, and concentration measurements. To account for the unknown magnitude of these uncertainties,
the coefficients of these relationships, $\sigma_{C,a}$, $\sigma_{C,b}$, *and* $\sigma_\delta$, were inferred alongside the model parameters.
The marginal posteriors of individual parameters were compared with their priors to evaluate whether observational data
provided information about these parameters, and whether this information was in conflict with the priors. In addition, two-
dimensional marginals were examined to identify potential identifiability issues. Finally, uncertainty in the model results was
calculated by propagating parameter uncertainty to the model results under consideration of their uncertainty for given
parameter values as formulated in the likelihood function:



$$f_{\text{post}}(\text{C}) = \int f_L(C|\theta)\, f_{\text{post}}(\theta)\, d\theta \qquad (6)$$

For the parameters with marginal posteriors in conflict with prior information, we conducted additional scenario analyses,
fixing parameters, and narrowing or widening prior distributions. These analyses evaluated the model's compatibility with
observational data if parameters better aligned with prior information, and assessed changes in posterior distribution with
weaker priors. These scenario analyses complemented the assessment of bioturbation uncertainty mentioned above.

**2.6 Discretization and numerical algorithms**

The partial differential equations outlined in Appendix B: *Reaction-diffusion model* were solved using the Method of Lines.
For spatial discretization, a grid was employed with cell thickness increasing progressively from the sediment surface toward
deeper layers. This adaptive grid design reduced the total number of cells required, while still maintaining high resolution
near the sediment-water interface, where steep concentration gradients typically occur (Appendix D: *Model discretization*).
The resulting system of ordinary differential equations (ODE) was solved by a standard ODE solver. Parameter inference
was conducted using two advanced Bayesian inference algorithms: Metropolis (Andrieu et al., 2003; Vihola, 2012) and
Hamiltonian Monte Carlo (Betancourt, 2017; Neal, 2011) algorithms.

**2.7 Model implementation**

The model was implemented in Julia (Bezanson et al., 2017) (https://julialang.org) to achieve high-performance and
facilitate automatic differentiation. The DifferentialEquations.jl package (Rackauckas and Nie, 2017) was used to solve the
system of ODEs; performance testing of several ODE solvers identified the FBDF solver (adaptive order and adaptive time-
step backward-differencing solver) as the most suitable for handling the stiffness of the ODE system. The ForwardDiff.jl
package (Revels et al., 2016) was used for automatic differentiation; Bayesian inference was conducted using the adaptive
Metropolis sampler from the AdaptiveMCMC package (Vihola, 2020), and the Hamiltonian Monte Carlo algorithm
implemented in the AdvancedHMC.jl package (Xu et al., 2020). Further implementation details are provided in Appendix E:
*Model implementation*. Simulations were performed at sciCORE (https://scicore.unibas.ch), the scientific computing centre
at the University of Basel.

**3. Sample collection and analyses**

**3.1 DIN concentrations and isotopes**

Sediment cores were retrieved at the deepest location of the Kreuztrichter basin in Lake Lucerne, a large oligotrophic lake in
Switzerland (Baumann et al., 2024), in April 2021 using a gravity corer with PVC liners. The sediment cores were stored at
4 °C and processed using two porewater-sampling methods: whole-core squeezing (WCS; (Bender et al., 1987)) for $NO_3^-$
samples, and Rhizon samplers (Rhizosphere research products, Wagenigen, NL) for $NH_4^+$ samples. The WCS technique



provides a high depth resolution near the sediment-water interface (0-5 cm, resolution: ~ 0.7-1 mm), where $NO_3^-$ is present
in porewaters, while the Rhizon sampling method allows collecting samples at greater sediment depths (> 5 cm, resolution: ≥
0.5 cm). $NO_3^-$ and $NH_4^+$ concentrations were measured using ion chromatography (940 Professional IC Vario, Metrohm).
$\delta^{15}N$-$NO_3^-$ and $\delta^{15}N$-$NH_4^+$ were determined using the denitrifier method (Casciotti et al., 2002; Sigman et al., 2001), and the
hypobromite-azide method (Zhang et al., 2007), respectively. In both methods, sample N from $NO_3^-$ or $NH_4^+$ is converted
into $N_2O$, which is then purified and analysed by isotope ratio mass spectrometry (Delta V Plus, Thermo Fisher Scientific).
The typical analytical precision is ~ 0.25‰ (McIlvin and Casciotti, 2010).

**3.2 Process rate measurements**

For model parameterization, reaction rates for denitrification, DNRA, and anammox were determined using established
protocols for $^{15}N$-tracer incubations (Holtappels et al., 2011). After recovery and sectioning of the core into 1-cm intervals, 1
g of sediment was placed into 12 mL gas-tight glass vials (Exetainers®, Labo, UK). These Exetainers were then filled with
anoxic, sterilized bottom water, amended with the following tracers: (Exp1) $^{15}NO_3^-$, (Exp2) $^{15}NH_4^+ + ^{14}NO_2^-$. Exetainers were
incubated at 6 °C in the dark, and terminated at designated time points (0, 6, 12, 24, and 36 hours) by adding $ZnCl_2$. Gas
headspace samples were analysed for the production of $^{14}N^{15}N$ and $^{15}N^{15}N$ using gas-chromatography isotope ratio mass
spectrometry (GC-IRMS; Isoprime, Manchester, UK). Linear regression of $^{14}N^{15}N$ and $^{15}N^{15}N$ production over time was
used to calculate $N_2$ production rates, with standard errors derived from deviations in the regression slopes across the five-
time points. For the determination of $^{15}NH_4^+$ production from $^{15}NO_3^-$ additions, $^{15}NH_4^+$ was chemically converted to $N_2$ gas
using the alkaline-hypobromite method (Jensen et al., 2011). The resulting $^{14}N^{15}N$ was quantified by GC-IRMS. Linear
regression of $^{14}N^{15}N$ production over time was used to calculate potential rates of $^{29}N_2$ (i.e., $^{15}NH_4^+$) production. Rates of
denitrification, DNRA, and anammox were calculated according to Holtappels et al. (2011) and Risgaard-Petersen et al.
(2003). Only data from the upper 1 cm were used to parameterize the model, as the investigated sediments displayed a
shallow nitracline and the highest anammox contribution at 0-0.5 cm depth.

**4. Results and Discussion**

The developed diagenetic N isotope model addresses existing knowledge gaps in understanding porewater N dynamics, and
aims to clarify the roles of distinct N-transformation processes in shaping the distribution of N isotopes to be potentially used
to constrain benthic N (isotope) fluxes across different environments. Here, we present (1) the results of Bayesian inference
applied to a large number (~ 60) of model parameters (see prior definition in Appendix C: *Prior values for inference*), with a
focus on assessing their uncertainty, (2) a detailed scenario analysis, focusing on parameters that exhibit significant shifts in
their marginal posterior distributions relative to their prior, as well as on the effect of variable contributions from different
$NO_3^-$ and $NO_2^-$ reduction pathways, and the impact of enhanced bioturbation on model outcomes, (3) a sensitivity analysis,
evaluating the importance of individual model processes in shaping benthic N isotope dynamics, (4) the importance of





process coupling in benthic N cycling, with a particular focus on the role of intermediate $NO_2^-$ in influencing $\delta^{15}N$-$NO_3^-$
dynamics. All results are based on porewater concentration, isotope, and rate measurement data from a sampling campaign
conducted in Lake Lucerne in April 2021. Additionally, we performed (5) a sensitivity analysis examining model output
responses to modifications of selected parameters using artificially simulated settings (e.g., variable contributions of
denitrification/anammox/DNRA); this analysis demonstrates the model's capability for addressing diverse research
questions.

### 4.1 Bayesian inference

The model implementation was highly efficient, achieving simulation times of about 12 s on an 13th Gen Intel® Core™ i9-
13,900K processor with 3.00 GHz and 64 GB of memory (of which only a small fraction was needed) for a 100-day
simulation starting from constant concentration profiles. This efficiency enabled the execution of Markov chains of 20,000
iterations within a few days on the scientific computing centre at the University of Basel (https://scicore.unibas.ch). By
combining these chains, samples of 100,000 iterations were generated. The Hamiltonian Monte Carlo algorithm
outperformed the adaptive Metropolis algorithm during burn-in to the core of the posterior distribution. However, for final
posterior sampling with about 60 parameters, adaptive Metropolis sampling proved more efficient in terms of effective
sample size per unit of simulation time. Despite these efforts in getting computational efficiency, and the use of advanced
MCMC algorithms, reaching convergence of the Markov chains remained challenging. We got five consistent Markov
chains without discernible trends for each scenario; however, some widening of the chains and the resulting effective sample
size on the order of 500 indicate that we are not able to get a good coverage of the tails of the posterior distribution. This
outcome demonstrates that incorporating so many uncertain model parameters pushes the limits of Bayesian inference in
terms of numerical tractability. However, the resulting uncertainty estimates are certainly more realistic than those obtained
by fixing many poorly constrained parameters to unique values to reduce the dimension of the parameter space.
The simulation results of solute concentration and $\delta^{15}N$ profiles in the most plausible Base scenario (Fig. 2) integrate prior
knowledge (Appendix C: *Prior values for inference*) with observational data through Bayesian inference. The profiles
closely reproduce the available, albeit limited, data, and conform to expected depth-related trends: oxidants (i.e., $O_2$, $NO_3^-$
and $SO_4^{2-}$) are readily consumed via aerobic mineralization and nitrification ($O_2$), denitrification ($NO_3^-$), and $SO_4^{2-}$ reduction.
While mineralization is assumed to involve negligible N isotopic fractionation, the first step of nitrification causes
significant enrichment in $^{15}N$ of the residual $NH_4^+$ pool, yielding $\delta^{15}N$-$NH_4^+$ values up to 11.2‰ at 0.15 cm, due to strong N
isotope fractionation, estimated at $\varepsilon_{Nit1} = 12.0‰$ (to $NO_2^-$) and 36.4‰ (to $N_2O$). Unfortunately, extremely low $NH_4^+$
concentrations measured in the top 2 cm hindered the determination and verification of the modelled $\delta^{15}N$-$NH_4^+$ in this zone
with field data. Both $NO_2^-$ and $N_2O$ accumulate in the upper 0.5 cm, reaching up to 0.4 μM and 2 μM, respectively. Below
0.3 cm, denitrification leads to the progressive $^{15}N$ enrichment of $NO_3^-$, $NO_2^-$ and $N_2O$, while $N_2$-producing mechanisms (i.e.,
denitrification and anammox) cause only minimal changes to the modelled $\delta^{15}N$-$N_2$ profile, due to the dominance of a large





pre-existing $N_2$ pool. For concentrations, the 95% credibility intervals of parametric uncertainty are rather narrow, whereas
the much broader total uncertainty is dominated by the lumped uncertainty term in the likelihood function, which primarily
reflects the model's structural uncertainty. The error, beyond the parameter error, is parameterized using the two sigma
values ($\sigma_{C,a}$ and $\sigma_{C,b}$; see Sect. 2.5), and exceeds what would arise from measurement and sampling alone. This suggests that
the larger error is attributable to the model's structural limitations. Conversely, $\delta^{15}N$ profiles exhibit small total uncertainty,
as model results for $\delta^{15}N$ closely match observational data, with minimal random and systematic deviations (parameterized
using the sigma value $\sigma_\delta$, see Sect. 2.5).
The model provides insights into the underlying process rates (Fig. 3) that shape the simulated profiles (Fig. 2). Vertical
profiles of transformation rates for $NH_4^+$, $NO_3^-$, $NO_2^-$ and $N_2O$ clearly illustrate the sequential dominance of different N-
transformation processes with increasing sediment depth and decreasing $O_2$ availability. Aerobic processes, namely aerobic
mineralization and nitrification, primarily control $NH_4^+$ transformation rates, peaking at 450 and 350 µM d$^{-1}$, respectively
(Fig. 3a). Nitrification sustains denitrification by producing both $NO_2^-$ (up to 350 µM d$^{-1}$) and $NO_3^-$ (up to 275 µM d$^{-1}$) in the
upper 0.4 cm (Fig. 3b-c). A strong spatial overlap of nitrification and denitrification emerges in the depth distribution of
processes affecting the $NO_2^-$ pool, suggesting a potential interplay between these pathways (Fig. 3c).
A key strength of this model is the incorporation of $N_2O$ as a state variable. Our model results reveal that, although $N_2O$
production via nitrification is minimal (not visible in Fig. 3d), the strong isotopic fractionation associated with this reaction
($\varepsilon_{Nit1,N2O}$ = 36.4‰) generates $N_2O$ with $\delta^{15}N$ values of -1.2‰ to –2.2‰ in the top 0.2 cm (Fig. 2c). At a depth of
approximately 0.35 cm, up to 2.1 µM of $N_2O$ accumulate, coinciding with the highest rates of $N_2O$ production through
denitrification. Conversely, $N_2O$ consumption by the last denitrification step peaks at 0.5 cm, leading to a progressive
increase in $\delta^{15}N$-$N_2O$ with depth. This zonation likely reflects the $O_2$ sensitivity of the distinct $N_2O$-producing and -
consuming processes. Specifically, $N_2O$ reductases are known to be strongly inhibited by $O_2$, and therefore exhibit greater
activity below the oxycline (Wenk et al., 2016). Although the model does not explicitly include the enzymes responsible for
N-transformation pathways, the chosen and estimated kinetic parameters reflect substrate affinity and inhibition strength.
Consequently, inhibition constants like $K_{O2,Den2}$ and $K_{O2,Den3}$ provide indirect insights into the $O_2$ dependency of these
enzyme-mediated reactions, effectively shaping the modelled redox zonation.
The model adequately captures the concentration and isotopic composition of the state variables, in agreement with field
measurement and the expected patterns of underlying N-transformation processes and reaction coupling (Fig. 2 and 3). One
key strength of the step-wise model is its ability to quantify reaction coupling, which is challenging to infer directly from
state variable pools (i.e., reactive intermediates), if they are rapidly turned over.



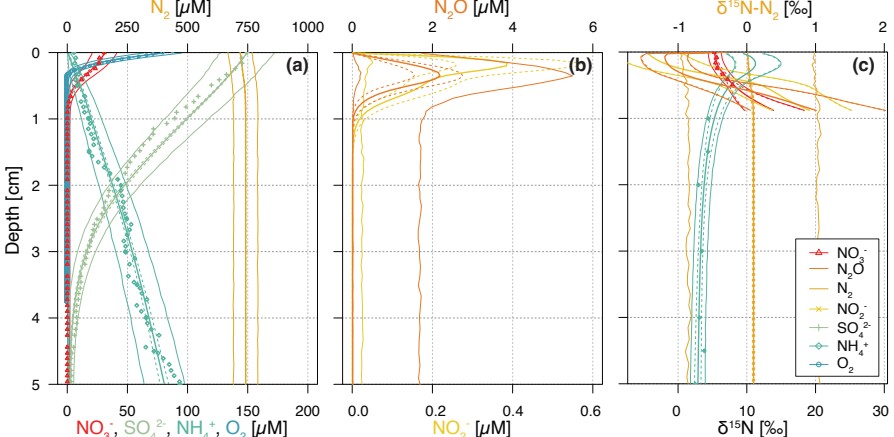

**Figure 2. Vertical porewater profiles of concentrations (a-b) and isotopic composition ($\delta^{15}$N) (c) of the state variables for the Base scenario. Continuous lines represent model simulations, while symbols represent observational data from Lake Lucerne. For $NH_4^+$ concentrations, filled diamonds represent low-resolution data from Rhizon sampling, while open diamonds represent the high-resolution WCS data, adjusted to align with absolute concentrations measured in the low-resolution dataset. Dashed lines enclose 95% credibility intervals resulting from parametric uncertainty, while thin solid lines represent total uncertainty.**

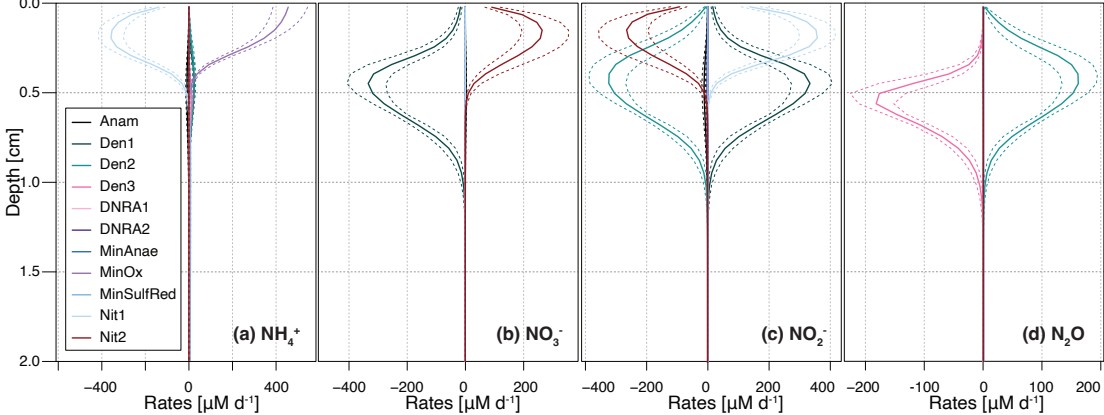

**Figure 3. Vertical profiles of transformation rates for distinct N-cycling processes affecting the $NH_4^+$, $NO_3^-$, $NO_2^-$, and $N_2O$ pools. Dashed lines enclose 95% credibility intervals resulting from parametric uncertainty. Positive reaction rate values indicate production, negative values indicate consumption of a given DIN species.**

To address the variable ranges for the model parameters found in the literature, and to reduce structural uncertainty imposed by fixed parameter values, we estimated a large set of parameters using Bayesian inference. The obtained joint posterior distribution of model parameters enabled us to assess the knowledge acquired from data. Marginal posterior distributions of individual parameters, and two-dimensional marginal distributions of parameter pairs, were particularly useful in this context (Fig. 4 shows examples for the four categories defined below; Fig. S1 provides an overview of all marginal prior and posterior parameter distributions). By comparing marginal posterior distributions with their corresponding priors, parameters were classified as well identifiable or poorly identifiable. While this classification involves some subjectivity in determining



how much narrower a posterior distribution should be compared to its prior distribution to classify such parameter as well identifiable, some clear patterns emerged:

1. Well identifiable parameters: The marginal posterior distribution is clearly narrower than the prior, indicating that data provide meaningful information about the parameter's value. Two cases were observed:

   a. The marginal posterior distribution is within the prior range, suggesting that the information from the data is in agreement with prior knowledge (Fig. 4a). Examples include: f factors for anammox ($f_{Anam,Den2}$ = 0.2) and both DNRA steps ($f_{DNRA1,Den1}$ = 0.005, $f_{DNRA2,Den2}$ = 0.005), estimated using $^{15}$N-tracer incubation experiments for the investigated system, and parameters such as $K_{NO3,Den1}$ and $K_{O2,MinOx}$, constrained from clearly defined oxidant declines. Maximum conversion rates for aerobic mineralization, denitrification, $SO_4^{2-}$ reduction, and anaerobic mineralization, as well as the $NH_4^+$ flux from deeper sediment layers, also belong to this category, although we approximated very wide priors by uniform priors (see Sect. 2.4), making it less visible in the plot.

   b. The marginal posterior distribution significantly deviated from the prior range (Fig. 4b), suggesting that the information from the data is in conflict with prior knowledge. The most striking example is $\varepsilon_{Den1}$, estimated at 2.8±1.1‰ for the Lake Lucerne dataset, far lower than the typical 15-25‰ reported in the literature for $NO_3^-$ reduction (Lehmann et al., 2003; Rooze and Meile, 2016), suggesting a reduced N-isotopic fractionation (or at least, of its expression) at the porewater level. This finding contrasts with model-derived values for the cellular isotope effect of $NO_3^-$ reduction observed in the porewater of marine sediments ($\varepsilon_{Den}$ > 10‰) (Lehmann et al., 2007). While a detailed investigation of the biological mechanisms behind such reduced expression across benthic environments is beyond the scope of this study and will be addressed separately by the authors, the potential role of reaction couplings in modulating benthic N isotope dynamics is discussed in Section 4.4.

2. Poorly identifiable parameters: The marginal posterior distribution resembles the prior distribution, suggesting poor identifiability. This can occur for two possible reasons:

   a. The parameter exerts negligible influence on the model output that corresponds to observational data (Fig. 4c). For example, parameters like the $N_2O$ yield during nitrification, $a_{N2O, Nit1}$ and $b_{N2O, Nit1}$, could not be constrained without specific data on $N_2O$ production. The current model encompasses several processes and state variables, which, at times, were hard to corroborate with the limited dataset in hand (a situation that may apply regularly to environmental studies, particularly in benthic environments). Therefore, their values were taken from previous studies (Ji et al., 2018). For other parameters, such as $\gamma_{NH4,DNRA1}$ and $\gamma_{NH4,DNRA2}$, little knowledge was acquired from the data in hand, due to the relatively low maximum rates of DNRA compared to other processes. In such cases, the posterior distribution may remain close to the prior, not because the prior range was incorrect, but because the available data could not further constrain it.

   b. Although data are available and the model output is sensitive to the parameter, other parameters influence the output similarly. This leads to parameter correlation in the posterior distribution and reduces identifiability, as



observed for $\gamma_{NH4,MinSulfRed}$ and $F_{NH4}$ (Fig. 4d), which exhibit correlation, making their estimates interdependent
(Guillaume et al., 2019). Here, the estimate of the $NH_4^+$ flux from the lower boundary of the model depends on
the estimate of the amount of $NH_4^+$ released via OM mineralization coupled to $SO_4^{2-}$ reduction.
The comparison of marginal priors and posteriors of the parameters (Fig. S1) demonstrates that excellent agreement between
model outputs and observational data (Fig. 2) can be achieved for 54 of the 58 estimated parameters compatible with their
priors. Exceptions include: the higher-than-expected rate for the second denitrification step relative to the first (expressed by
the factor $f_{Den2,Den1}$), the large half-saturation constant for $SO_4^{2-}$ reduction ($K_{SO4,MinSulfRed}$), and smaller-than-expected N
isotope effects for the first steps of denitrification and nitrification ($\varepsilon_{Den1}$ and $\varepsilon_{Nit1,NO2}$, respectively). The largest deviation is
observed for $\varepsilon_{Den1}$, which is further examined in the next subsection.
Notably, the seven parameters, for which a uniform prior was chosen to approximate a very wide prior ($k_{MinOx}$ $k_{Den1}$,
$k_{MinSulfRed}$, $k_{MinAnae}$, $k_{Nit1}$, $F_{NH4}$, $\delta^{15}N,F_{NH4}$), were identifiable, indicating that highly system-specific prior knowledge is not
crucial for these estimates. Most of the other model parameters showed limited narrowing of the marginal posterior relative
to the prior, reflecting the rather limited information gain that can be obtained from data. The three model error parameters
($\sigma_{C,a}$, $\sigma_{C,b}$, $\sigma_{\delta}$) were well identifiable and will be used in the following sections to compare the fit quality across different
modelling scenarios.

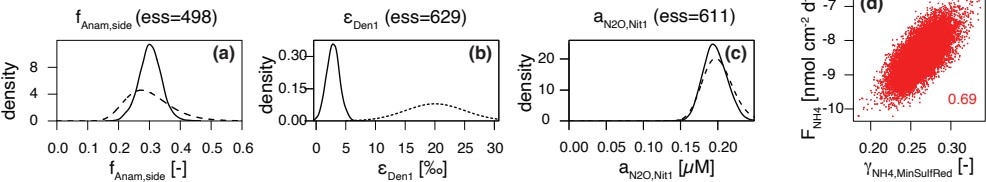


**Figure 4. Prior (dashed line) and posterior marginal distributions (continuous line) for illustrative parameters, which could be**
**identified and showed (a) good ($f_{Anam,side}$) and (b) poor agreement ($\varepsilon_{Den1}$) with prior knowledge, and (c) for parameters, that could**
**not be identified ($a_{N2O, Nit1}$); 2D correlation plot for $\gamma_{NH4,MinSulfRed}$ versus $F_{NH4}$ (d).**
**4.2 Scenario analysis**
Building on the findings discussed in the previous subsection, we explored the apparent prior-data conflict regarding $\varepsilon_{Den1}$ in
greater detail. Additionally, we assessed whether the estimated process rates overlooked potential reaction coupling, which
might go undetected through $^{15}$N-tracer incubation experiments, by exploring the variability in contributions of anammox
and DNRA (i.e., $f_{Anam}$, $f_{DNRA1}$ and $f_{DNRA2}$). Lastly, given the uncertainty regarding solute-diffusion enhancement by
bioturbation, we investigated a scenario with increased bioturbation. These considerations led to four key scenarios:
A.  *Narrow priors for $\varepsilon$.* This scenario investigated the effects of restricting $\varepsilon$ variability to a narrower range (prior

standard deviation of 1‰ instead of 5‰). The aim was to test whether the marked reduction in the marginal



500   posterior of $\varepsilon_{Den1}$ persisted under stricter prior assumptions, and whether this decreased flexibility significantly

501   impacted the quality of the model fit.

502  B. *Fixed $\varepsilon$.* Here, the model output was assessed under the assumption that the literature data regarding N isotope

503   effects are correct (i.e., $\varepsilon$ values not estimated). This scenario complemented Scenario A by testing whether a good

504   fit to the data could still be achieved by fixing the $\varepsilon_{Den1}$ value (and all other isotope effects) at its prior mean.

505  C. *Wider priors for f.* In this scenario, greater variability in DNRA and anammox contributions (prior standard

506   deviation of 100% instead of 25%) was allowed to test the impact of relaxed prior assumptions on the relative

507   contributions of these processes in the model output.

508  D. *Enhanced bioturbation.* This scenario simulated a faster solute-diffusive transport due to higher infaunal activity by

509   doubling the bioturbation coefficient ($D_{bio} = 2$ cm$^2$ d$^{-1}$ instead of 1 cm$^2$ d$^{-1}$), to investigate the sensitivity of the

510   results to this uncertain parameter, which was not included in the Bayesian analysis. In the model, the bioturbation

511   strength at the sediment surface is defined by the parameter $D_{bio}$, and it decreases exponentially with depth, with the

512   typical bioturbation depth parameter, *depth$_{bio}$*.

The results demonstrate a strong dependence of the estimated parameters on the chosen prior assumptions (Fig. 5). Across all
scenarios, marginal posterior distributions for the selected parameters are generally narrower than the prior distributions,
though results vary substantially. In Scenario A (*Narrow priors for $\varepsilon$*), restricting the prior range significantly constrained
$\varepsilon_{Den1}$, limiting its deviation from the prior (Fig. 5m; note that the prior for Scenario A is five times narrower than the one
shown, which represents the prior for all other scenarios). These results closely resemble those from Scenario B (*Fixed $\varepsilon$*),
where no deviation was possible (Fig. 5, Fig. S2). Both scenarios exhibit lower denitrification rates than the Base scenario
(Fig. 5b), but comparable fit quality for total ($^{14}$N + $^{15}$N) concentration, quantified by $\sigma_{C,a}$ (i.e., the dominant term of standard
deviation of the model error for concentrations, see Sect. 2.5) (Fig. 5x). On the other hand, Scenarios A and B display poorer
fit quality for $\delta^{15}$N profiles, indicated by a large value of $\sigma_\delta$ (Fig. 5z), suggesting that the model structure cannot adequately
reproduce the $\delta^{15}$N-NO$_3^-$ profiles without adapting the $\varepsilon_{Den1}$ value. While biological isotope effects of 15-30‰ are typical for
NO$_3^-$ reduction (Lehmann et al., 2007), lower values under almost-complete NO$_3^-$ consumption have been reported (Thunell
et al., 2004; Wenk et al., 2014). This finding is further confirmed by comparable marginal posteriors for $\varepsilon_{Den1}$ across all
scenarios considered in this study, besides scenarios A and B.
In Scenario C (*Wider f*), allowing greater variability in anammox and DNRA contributions results in the lowest $f_{Anam,Den2}$
values, although such deviation is not substantial compared to the Base scenario output (Fig. 5i). The estimated $f_{DNRA1,Den1}$
and $f_{DNRA2,Den2}$ values in Scenario C mostly align with those of the Base scenario, corroborating the marginal role of DNRA
in Lake Lucerne. Such findings confirm the accuracy of the rate measurements performed with $^{15}$N tracer incubations.
Scenario D (*Enhanced bioturbation*) stands out with the highest conversion rates (i.e., $k_{MinOx}$, $k_{MinSulfRed}$, and $k_{Nit1}$) (Fig. 5a,e,g)
to ensure sufficient oxidant consumption at higher supply/flux rates (reproducing the observed gradient despite higher





diffusivity). Despite these changes, bioturbation had negligible effects on porewater N isotope dynamics, with estimated
isotope effects and fit quality for $\delta^{15}N$ profiles ($\sigma_\delta$) comparable to those of the Base scenario.
The obtained concentration depth profiles for the four scenarios are generally comparable, as newly estimated parameters
ensured good fitting of the data (Fig. S2). However, in Scenarios A and B, stricter constraints on prior knowledge for
parameter estimation result in little to no suppression of all isotope effects (i.e., relatively strong N isotopic fractionation),
leading to great variability in the $\delta^{15}N$ profiles. Poor fits to the $\delta^{15}N$ data are observed under these conditions, as evidenced
by the greater $^{15}N$ enrichment of the $NO_3^-$ pool compared to the measured-data profiles (Fig. S2). Similarly, the $\delta^{15}N$-$N_2O$
profiles exhibit sharp declines to approximately -15‰ in the upper 0.5 cm under Scenarios A and B, driven by the strong
expression of $\varepsilon_{Nit1,N2O}$ (40.1‰ and 40.0‰, respectively). In contrast, Scenarios C and D closely resemble the Base scenario,
with only minor $\delta^{15}N$-$N_2O$ variations.





**Figure 5. Marginal probability densities across the five considered scenarios for selected estimated parameters, showing both prior (dashed line) and posterior distributions (continuous lines):** *Base scenario* (SD$_f$ = 25%, SD$_\varepsilon$ = 5‰, $D_{bio}$ = 1 cm$^2$ d$^{-1}$), *Narrower $\varepsilon$* (SD$_\varepsilon$ = 1‰), *Fixed $\varepsilon$* (i.e., $\varepsilon$ taken from bibliography), *Wider f* (SD$_f$ = 100%) and *Enhanced bioturbation* ($D_{bio}$ = 2.0 cm$^2$ d$^{-1}$). Of the ~ 60 estimated parameters, those shown here were selected for their relevance to the discussion. See main text for further details.

**4.3 Importance of modelled processes and their impact on porewater N isotope signatures**

The importance of modelled processes and their impact on N isotope signatures were investigated by selectively deactivating individual processes and comparing the model outputs to the Base scenario. Aerobic mineralization, denitrification, and SO$_4^{2-}$ reduction were considered essential to preserve redox zonation (e.g., sequential decline of O$_2$, NO$_3^-$, and SO$_4^{2-}$) and N dynamics. The following processes were individually turned off: (a) nitrification ("NitOff"); (b) anammox ("AnamOff"); and (c) DNRA ("DNRAOff"). Initially, each process was simply inactivated to assess its impact on model outputs (Fig. 6).



Subsequently, inference was conducted after deactivating each process, to investigate their importance for model
performance, parameter and flux estimation, and for the identifiability of rate parameters by evaluating the quality of the fit
to the data, especially on the $\delta^{15}$N profiles (Fig. 7, Fig. S3, Fig. S4).
Switching off nitrification significantly alters the model output compared to the Base scenario (Fig. 6a-b,e-f), indicating its
central role in the benthic N dynamics. Key effects include $NH_4^+$ accumulation throughout the investigated depths, with a
flattening of the $\delta^{15}$N-$NH_4^+$ profile (i.e., less curvature towards higher $\delta^{15}$N values) in the upper 0.5 cm, as the only other
source of $^{15}$N-enriched $NH_4^+$ besides nitrification would be anammox, which is inhibited under oxic conditions. Furthermore,
nitrification-denitrification coupling via $NO_2^-$ weakens in this scenario, resulting in lower overall $N_2$ production (as indicated
by the lower maximum $N_2$ concentration of 734 µM compared to 745 µM in the Base scenario). These results suggest that
partially reducing, or fully eliminating, nitrification lowers the system's capacity to act as an efficient N sink. In other words,
the findings confirm that nitrification is a critical process that, when closely coupled to denitrification, helps to enhance the
ecosystem's potential to remove fixed N. All other N-isotopic state variables also show a flatter $\delta^{15}$N profile, with only a
progressive enrichment in $^{15}$N below 0.5 cm, primarily driven by denitrification ($NO_3^-$, $NO_2^-$, and $N_2O$). The impact of
disabling nitrification is clearly reflected in the $\delta^{15}$N-$N_2O$ profile across the upper 0.3 cm, where the typical nitrification-
induced dip is absent, and $\delta^{15}$N-$N_2O$ values remain relatively constant (~7-8‰). In contrast, the effects of turning off
anammox or DNRA are more subtle, owing to their generally lower reaction rates in Lake Lucerne (Fig. 6c-d,g-h). Notably,
in the absence of anammox, $N_2O$ exhibits lower $\delta^{15}$N values in the upper 0.3 cm compared to the Base scenario, likely due to
higher $N_2O$ yields via nitrification, as reduced competition for $NH_4^+$ with anammox provides more substrate for nitrification.
Upon running inference for each case, concentration and N isotope profiles for the NitOff, AnamOff, and DNRAOff
scenarios are generally similar to those of the Base scenario (Fig. S3), with notable exceptions in the NitOff case. In the
absence of nitrification, $NH_4^+$ accumulates and the $\delta^{15}$N-$NH_4^+$ profile remains largely flat, since anammox, the only other
$NH_4^+$-consuming process, is minimal under oxic conditions. No $\delta^{15}$N-$NH_4^+$ measurements are available for the top 1 cm, so
the model output could not be verified with field data. The $N_2O$ pool systematics also diverge between the NitOff and Base
scenarios. Specifically, in the NitOff case, no nitrification-derived $N_2O$ accumulates in the upper 0.4 cm, and consequently,
the $\delta^{15}$N-$N_2O$ profiles lacks the typical nitrification-associated decline in this layer. Instead, $N_2O$ becomes progressively
enriched in $^{15}$N below 0.4 cm. While most estimated parameters and fluxes are consistent across the four scenarios, the
NitOff scenario stands out again, exhibiting strong effects on the anammox rates and associated isotope effects (e.g.,
$f_{Anam,Den2}$, $\varepsilon_{Anam,NH4}$) (Fig. S4), as well as on benthic fluxes of $NH_4^+$, $NO_2^-$, $NO_3^-$ and $N_2O$ (Fig. 7). Nonetheless, the $NH_4^+$
concentration profile is well-captured, as indicated by a low $\sigma_{C,a}$, reflecting a good match between model and concentration
data even in the absence of nitrification. This finding implies that the model cannot resolve the relative contributions of
nitrification versus anammox to $NH_4^+$ consumption based on the concentration and isotope data, highlighting the importance
of prior knowledge regarding $f_{Anam,Den2}$.



The comparison of process rates across these four scenarios provides insights, unveiling the extent of process coupling and
competition (Fig. S5) (Hines et al., 2012). For instance, anammox and nitrification compete for both $NH_4^+$ and $NO_2^-$ as
substrates, causing the rate of one process to be enhanced, when the other is switched off. For instance, $NH_4^+$ oxidation and
$NO_2^-$ production rates via nitrification (Nit1) are higher (~ 0.2 cm depth) in the AnamOff scenario than in the Base scenario.
Even more obviously, enhanced rates of $NH_4^+$ oxidation, $NO_2^-$ consumption, and $NO_3^-$ production via anammox are observed
in the NitOff scenario than in the Base scenario. Process coupling, specifically nitrification-denitrification, is further
confirmed by lower rates for $NO_2^-$ reduction via denitrification (Den2) in the absence of nitrification. In general, the
influence of DNRA on production and consumption rates of the considered state variable appears minimal, owing to the
limited environmental relevance of DNRA in Lake Lucerne. Overall, the similarly good fits obtained across these three
scenarios and the *Base* scenario reflect the poor identifiability of the switched off processes; this suggests that the data can be
well-fitted even without these three processes, emphasizing the importance of prior knowledge about their environmental
relevance.

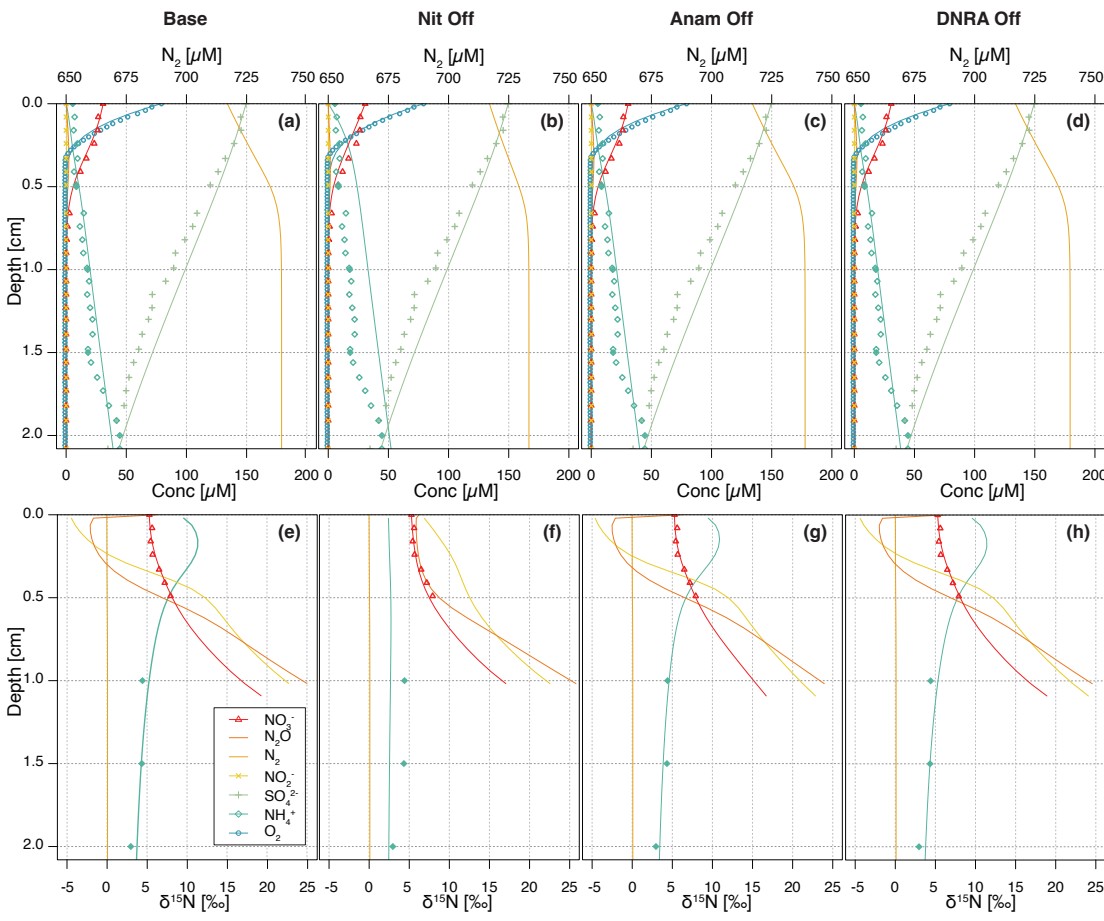


**Figure 6. Vertical concentration (a-d) and isotopic composition (e-h) profiles for state variables. Model output obtained with all**
**processes included (a, e) are compared with model simulations where individual processes are switched off: nitrification (b, f),**





anammox (c, g), and DNRA (d, h), without running inference again. Continuous lines represent the model output, while symbols represent measured data from Lake Lucerne. For $NH_4^+$, open diamonds represent the high-resolution dataset, adjusted to align with absolute concentrations measured in the low-resolution dataset (filled diamonds).

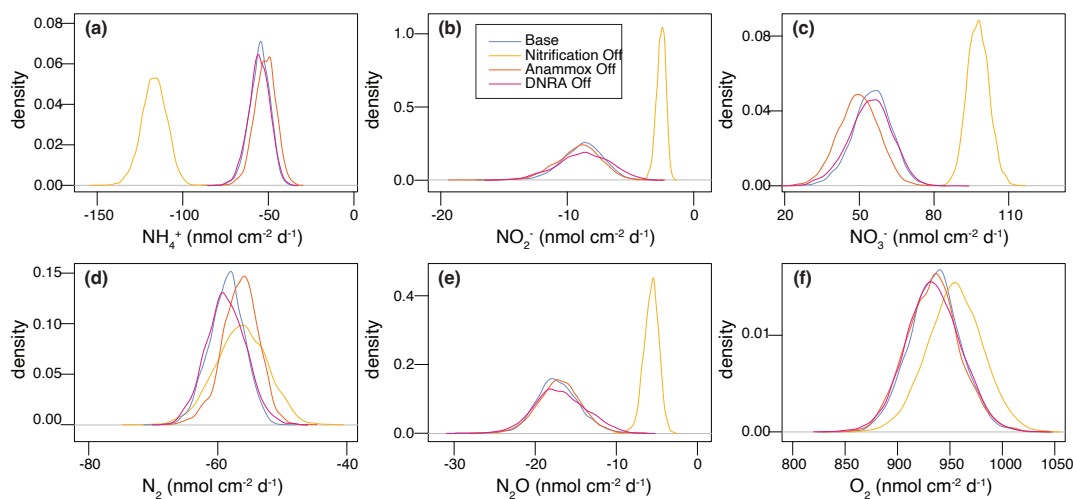

Figure 7. Posterior marginal probability distributions of modelled sediment-water interface fluxes (in nmol cm$^{-2}$ d$^{-1}$) for all state variables, generated from inference runs, across the four scenarios considered for model validation against experimental data from Lake Lucerne.

## 4.4 The role of process coupling via $NO_2^-$

Previous models of benthic N isotope dynamics have focused on individual reactions or overlooked the role of intermediate species, such as $NO_2^-$ (Kessler et al., 2014; Lehmann et al., 2007). Our study confirms that $NO_2^-$ plays a critical role in coupling multiple N-transformation processes and shaping benthic N isotope dynamics, including that of $\delta^{15}$N-$NO_3^-$. While such process coupling has been examined in the water column (Frey et al., 2014), it remains, to our knowledge, largely unexplored in sedimentary environments.

To assess the significance of this coupling, we implemented a one-step denitrification approach that bypasses $NO_2^-$ as an intermediate, replacing the three-step pathway used throughout this paper (Fig. 8). In this simplified model, $NO_2^-$ concentrations and isotopic signatures are shaped solely by nitrification (and to a marginal extent, DNRA and anammox), as denitrification no longer contributes to $NO_2^-$ production. This modification leads to significantly reduced $NO_2^-$ accumulation, restricted to the upper 0.3 cm, and lower anammox activity, due to a lack of $NO_2^-$ substrate below the oxycline. The absence of denitrification-derived $NO_2^-$ has profound effects on the N isotope dynamics. First, a consistent ~15‰ offset between $\delta^{15}$N-$NO_3^-$ and $\delta^{15}$N-$NO_2^-$ is evident across all modelled depths (Fig. 8c). This offset is ascribed to the isotope effect of the second nitrification step ($\varepsilon_{Nit2}$ = -13.7‰), and the lack of $^{15}$N enrichment in the $NO_2^-$ pool from denitrification. Second, the estimated isotope effect for $NO_3^-$ reduction ($\varepsilon_{Den}$) increases to 5.5±0.9‰, nearly double than in the Base scenario, indicating that elevated $\delta^{15}$N-$NO_3^-$ values in the field data may, to some extent, reflect $NO_2^-$ isotope dynamics, rather than solely the effect of $NO_3^-$ reduction (Fig. 1).



These findings emphasise the importance of both $NO_2^-$-producing and -consuming processes in modulating $\delta^{15}N$-$NO_3^-$, and
consequently, estimates of $\varepsilon_{Den1}$. Although nitrification is typically aerobic and denitrification anaerobic, evidence exists that
indicates spatial overlap of these two processes at the bottom of oxyclines in natural aquatic environments (Frey et al., 2014;
Granger and Wankel, 2016) at the bottom of the oxycline. In this transition zone, $NO_2^-$ produced by either pathway can be
oxidised to $NO_3^-$ or reduced to $N_2O$, $NH_4^+$ or $N_2$ (Fig. 3), significantly affecting its $\delta^{15}N$ signature (depending on the N-
branching). For instance, $NO_2^-$ reduction to $N_2O$ enriches the residual $NO_2^-$ pool in $^{15}N$. If this $^{15}N$-enriched $NO_2^-$ is
subsequently oxidized to $NO_3^-$ (a reaction that exhibits an inverse kinetic isotope effect), the resulting $NO_3^-$ will be markedly
enriched in $^{15}N$ (Fig. 1). Such interactions have been shown to influence apparent isotope effects for $NO_3^-$ in the water
column (Frey et al., 2014), and likely exert similar effects in sediments, where sharp redox gradients create overlapping
zones of nitrification and denitrification. This coupling may explain the discrepancy in estimated $\varepsilon_{Den1}$ values between the
Base scenario (2.8±1.1‰) and the one-step denitrification model approach (5.5±0.9‰).
Anammox further complicates these dynamics, as it depends on $NO_2^-$ excreted into the environment. Without denitrification,
which releases $NO_2^-$ (Sun et al., 2024), anammox is substrate limited (Fig. 8). Thus, while previous benthic studies estimated
denitrification isotope effects using one-step denitrification approaches (Lehmann et al., 2007), our findings call for the
adoption of a stepwise modelling approach (Sun et al., 2024) that better captures the interdependence of N-transformation
pathways, and their integrated effects on $NO_3^-$ isotope dynamics. A more detailed examination of these interactions is
essential for refining our understanding and quantification of isotope effects associated with $NO_3^-$ reduction in sedimentary
systems.

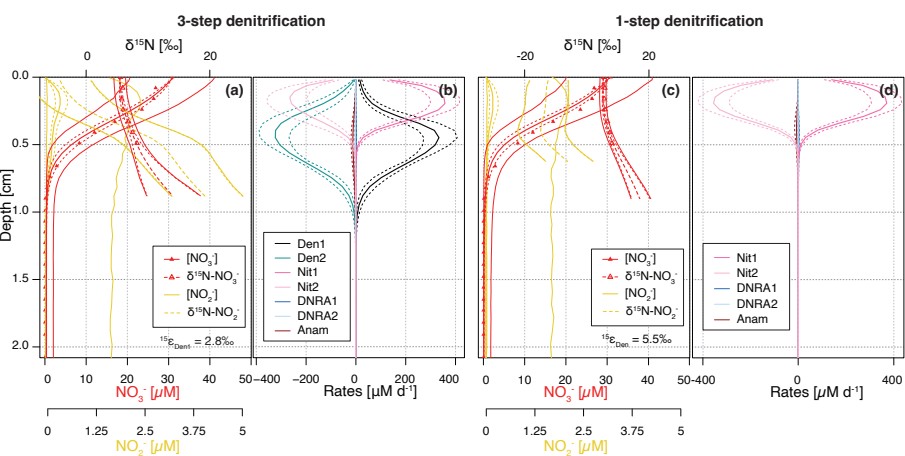

**Figure 8. Depth profiles of $NO_3^-$ and $NO_2^-$ concentrations and N isotopic composition (A,C), and rates of $NO_2^-$-producing and -**
**consuming processes (B,D), as simulated by the Base scenario (A,B), and the one-step denitrification approach (C,D). In the one-**
**step approach, $NO_3^-$ is reduced directly to $N_2$, omitting $NO_2^-$ as an intermediate; thus, no $NO_2^-$ is produced or consumed through**
**denitrification. Dashed lines enclose 95% credibility intervals resulting from parametric uncertainty.**





### 4.5 Model applicability in distinct scenarios

Beyond applying and testing the developed diagenetic N isotope model at our site of interest (Lake Lucerne), we believe its
strength hinges on its versatility to address distinct research questions and objectives. We explored two scenarios as
examples of how the model can be adapted to provide insights into the N cycle in benthic environments and the N isotopic
fingerprints that the combined N-cycling processes leave behind (Fig. 9). Understanding these fingerprints and how they
might be modulated in natural environments (e.g., through the variable balance between individual processes constrained by
environmental conditions) is important for correctly interpreting the distribution of $^{15}N/^{14}N$ ratios in N species as
biogeochemical tracer, helping to pinpoint and disentangle individual N-turnover processes where they co-occur.
For comparison purposes, we used the estimated parameters from the Base scenario and modified the relative importance of
$NO_3^-$ or $NO_2^-$ reduction via (i) denitrification vs. DNRA, and (ii) denitrification vs. anammox. This was done by
progressively increasing the factors that define the contributions of DNRA ($f_{DNRA1,Den1}$ and $f_{DNRA2,Den2}$) and anammox
($f_{Anam,Den2}$) from 0 (i.e., no DNRA/anammox) to 2 (corresponding to DNRA and anammox accounting for 2/3 of the total
$NO_3^-$ and $NO_2^-$ reduction, respectively). Simultaneously, the rates of the first two steps of denitrification ($k_{Den1}$ and $f_{Den2,Den1}$)
were adjusted to maintain consistent overall $NO_3^-$ and $NO_2^-$ reduction rates across scenarios. These model results were not
validated against observational data and should therefore be considered as illustrative examples of the model's sensitivity to
selected parameters, rather than as predictions with direct environmental relevance.
i.   N removal versus N retention
The model results confirm the spatial co-occurrence of DNRA and denitrification, with peak $NO_3^-$ (data not shown)
and $NO_2^-$ (Fig. 9a) reduction activities localized between 0.4-0.6 cm depth. In contrast, $NH_4^+$ and $N_2$ production
exhibit subtle differences in depth distribution: $NH_4^+$ production via DNRA extends across a broader sediment layer
than $N_2$ production via denitrification (Fig. 9b). This pattern likely reflects the inhibitory effect of $O_2$ on $N_2O$
reduction, the final denitrification step, pushing $N_2$ production to deeper, anoxic layers below the oxycline.
Reduction of $NO_3^-$ exhibits distinct isotope effects depending on the pathway: denitrification ($\varepsilon_{Den1} \approx 2.8\pm1.1‰$)
and DNRA ($\varepsilon_{DNRA1} \approx 20.0\pm2.9‰$), according to our model estimates (Fig. 5m,v). This large difference reflects the
difficulty of constraining DNRA isotope effects through Bayesian inference, due to its low environmental relevance
in the top 1 cm of Lake Lucerne sediments. Although not proven so far, this isotope offset implies that $NO_3^-$
reducers impart distinct isotopic fractionation depending on the pathway, which is rather implausible. However, if
true, increasing DNRA activity would lead to a stronger $^{15}N$ enrichment in the residual $NO_3^-$ pool (Fig. S6d), with
downstream impacts on the product pools ($N_2$ and $NH_4^+$) (Fig. 9c-d).
Denitrification-derived $N_2$ mixes with a large ambient $N_2$ pool (717 μM; $\delta^{15}N \sim 0$ ‰), resulting in slightly elevated
$\delta^{15}N$-$N_2$ values in the top 1 cm. While this increase is subtle ($\Delta\delta^{15}N < 0.1‰$), it becomes more pronounced as a
larger fraction of $NO_3^-$ (and subsequently $NO_2^-$) is reduced to $N_2$ (denitrification) rather than to $NH_4^+$ (DNRA) (Fig.
9c) due to the distinct isotope effects associated with $NO_3^-$ reduction via denitrification and DNRA. Under full

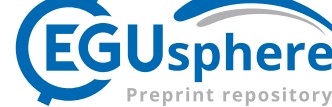

expression of the denitrification isotope effect (i.e., $\varepsilon_{Den1} \approx 20‰$), $\delta^{15}N\text{-}N_2$ much lower than 0‰ would be expected;

in contrast, $\varepsilon_{Den1} \approx 2.8‰$ likely suppresses such isotopic dynamics, resulting in only subtle $\delta^{15}N\text{-}N_2$ changes. As

more $NO_3^-$ is reduced via DNRA ($\varepsilon_{DNRA1} \approx 20.0‰$) than via denitrification ($\varepsilon_{Den1} \approx 2.8‰$), a stronger $^{15}N$ depletion

is expected in the $NO_2^-$ pool; if this $NO_2^-$ is then reduced to $N_2$ will lead to lower $\delta^{15}N\text{-}N_2$ than in a purely-

denitrifying case. Such interaction can explain the shift toward lower $\delta^{15}N\text{-}N_2$ values as $NO_3^-$ is increasingly

reduced via DNRA with a strong isotope effect recorded in our model. Thus, the slightly elevated $\delta^{15}N\text{-}N_2$ values

observed in our model confirms that denitrification dominates over DNRA, and operates with a reduced isotope

effect (2.8‰), likely due to diffusive limitation.

In contrast, enhanced DNRA activity leads to $NH_4^+$ accumulation and a progressive decrease in $\delta^{15}N\text{-}NH_4^+$ in the

upper 0.5 cm, consistent with strong isotopic fractionation during DNRA (Fig. 9d). This $NH_4^+$ pool appears to

promote nitrification, as indicated by higher $NH_4^+$ and $NO_2^-$ oxidation rates (Fig. S6a-b), resulting in the production

of $^{15}N$-depleted $NO_2^-$ (Fig. S6c). Notably, if this isotopically light $NO_2^-$ is subsequently reduced via denitrification,

it can lead to the formation of $N_2$ with unusually low $\delta^{15}N$ values, even if denitrification itself operates with a

modest isotope effect. This secondary effect underscores how DNRA not only alters substrate availability but also

indirectly influences the isotopic composition of denitrification end products. The strong spatial overlap of DNRA,

denitrification and nitrification highlights the central role of DNRA in fuelling internal N recycling (Wang et al.,

2020) with implications that extend to the $\delta^{15}N$ of both intermediate and terminal N pools.

Thus, if $NO_3^-$ reduction via DNRA and denitrification occurs with distinct isotope effects, our model has the

potential to disentangle their respective contributions based on $\delta^{15}N$ profiles of $NO_3^-$ and $NH_4^+$, and to a lesser

extent of $N_2$ and $NO_2^-$. Importantly, our results underscore a potentially critical, yet underappreciated, coupling

between DNRA and nitrification in benthic environments. If verified, this interaction, largely invisible in

concentration profiles alone, can significantly influence isotopic signatures and must be considered when

interpreting sediment N dynamics through an isotope lens.

ii.     N removal via denitrification versus anammox

The results for this case scenario reveal, somewhat unexpectedly, some similarities between denitrification and

anammox with respect to $NO_2^-$ reduction to $N_2$ and associated N isotope signatures. The isotope effects associated

with denitrification are low (2.8‰ for $NO_3^-$ reduction and 7.9‰ for $NO_2^-$ reduction), whereas anammox imparts

stronger isotopic fractionation (14.4‰ for $NO_2^-$ reduction to $N_2$ and -30.0‰ for its oxidation to $NO_3^-$). These values

reflect parameter estimations specific to Lake Lucerne's surface sediments (upper 1 cm), where anammox activity

is low.

Both $NO_2^-$ reduction and $N_2$ production peak around 0.5 cm depth, with minor differences in the thickness of the

active layer due to variations in substrate affinity between modelled processes (Fig. 9e-f). The total rate of $NO_2^-$

reduction to $N_2$, via either anammox or denitrification, remains consistent across all case scenarios. Nonetheless,





slight differences can be observed in some N pools as anammox becomes the dominant fixed-N loss path. Increased
anammox activity leads to elevated $N_2$ and $NO_2^-$ concentrations (Fig. 9g-h), likely due to the use of $NH_4^+$ as a
substrate, which mitigates substrate limitation under low $NO_2^-$ availability (i.e., 1.3 mol $NO_2^-$ needed to produce 1
mol $N_2$ via anammox versus 2 mol $NO_2^-$ via denitrification). When anammox prevails, $\delta^{15}N$-$NO_2^-$ values increase
due to the stronger isotope effect associated with $NO_2^-$ reduction via anammox relative to denitrification. This
enrichment is partially counterbalanced by the inverse kinetic isotope effect during $NO_2^-$ oxidation to $NO_3^-$ (Brunner
et al., 2013), leading to $^{15}N$-enriched $NO_3^-$ below 0.8 cm; notably, this isotopic shift occurs without significant
changes in total $NO_3^-$ concentrations (Fig. S6g-h). Lastly, substantial differences emerge in the $NH_4^+$ pool: higher
anammox activity correlates with lower $NH_4^+$ concentrations and elevated $\delta^{15}N$-$NH_4^+$ values throughout most of the
sampled depths (Fig. S6e-f). This isotopic enrichment likely overlaps with the effect of nitrification on the $NH_4^+$
pool in the upper 0.3 cm.
While some differentiation between denitrification and anammox is evident in the isotope signatures of $NO_3^-$ and
$NH_4^+$, the expected contrasts in the $NO_2^-$ and $N_2$ pools are surprisingly muted. This near-indistinguishability in
isotopic outcomes suggests a degree of functional and isotopic redundancy between the two pathways under the
modelled conditions. These results highlight the need for further investigation, particularly through refined isotope-
based methods (e.g., inclusion of $NO_x$ O-isotopes or clumped nitrate isotopes) and more mechanistic modelling, to
distinguish the respective contributions of denitrification and anammox to N removal in sedimentary systems.



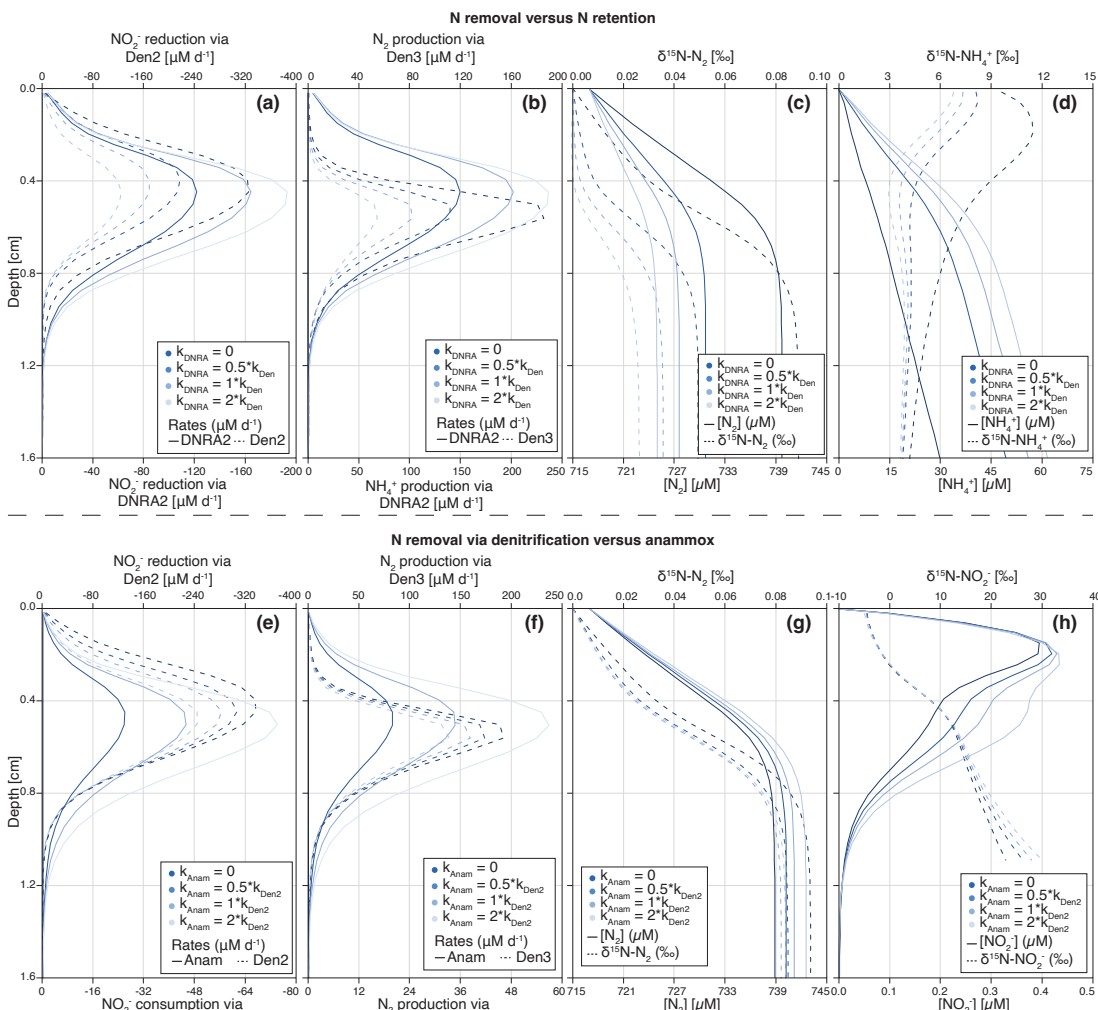


**Figure 9. Depth profiles of process rates, solute concentrations and δ$^{15}$N values for the two idealized case scenarios investigated: (i) NO$_3^-$ reduction via DNRA and denitrification (a-d), (ii) N$_2$ production via anammox and denitrification (e-h). Shadings represent different model scenarios within each case, as defined in the legend. For case (i), colour shading lightens with increasing contribution of DNRA (relative to denitrification) to total NO$_2^-$ reduction. DNRA accounts for 0‰ (f$_{DNRA}$ = 0), 33‰ (f$_{DNRA}$ = 0.5), 50% (f$_{DNRA}$ = 1) and 66% (f$_{DNRA}$ = 2) of total NO$_2^-$ reduction (panel a). The resulting effects on the production rates of NH$_4^+$ and N$_2$ (b), as well as on their concentrations and N isotopic composition (c-d), are shown. For case (ii), colour shading lightens with increasing contribution of anammox (relative to denitrification) to total NO$_2^-$ consumption and associated N$_2$ production. Anammox contributes 0‰ (f$_{Anam}$ = 0), 33‰ (f$_{Anam}$ = 0.5), 50% (f$_{Anam}$ = 1) and 66% (f$_{Anam}$ = 2) of total NO$_2^-$ consumption (e-f). The resulting impacts on N$_2$ and NO$_2^-$ concentrations and δ$^{15}$N values are shown in panels g-h.**

## 5. Conclusions

We developed a comprehensive diagenetic N isotope model that integrates multiple N transformations in benthic environments. The model's complexity requires the use of prior knowledge in addition to the observed data, in order to achieve the most plausible descriptions of the ongoing processes. To address uncertainty in prior knowledge, and to reduce



structural errors associated with fixed parameter values, we applied Bayesian inference for a large parameter set (~60) for
data analysis. The computational demands of this approach were met by implementing the model in Julia, with compatibility
for automatic differentiation to allow for advanced Markov chain Monte Carlo algorithms needed for Bayesian inference.
Despite these optimization efforts to enhance efficiency, inference runs still took 2-3 weeks of computation time (in addition
to preceding simulations to reduce burn-in) to achieve sufficiently good convergence of the Markov chains of the posterior
parameter distribution. Alongside concentrations and $\delta^{15}$N values for different N species, the model provides depth profiles
of process rates and all fluxes, including their uncertainties. These outputs enable a detailed assessment of the processes
shaping N cycling (i.e., concentration profiles) and isotope patterns in sediments.
Application of the developed model to a test dataset from Lake Lucerne successfully reproduced measured profiles of $O_2$,
$SO_4^{2-}$, $NH_4^+$, $NO_2^-$, $NO_3^-$, $\delta^{15}$N-$NH_4^+$, and $\delta^{15}$N-$NO_3^-$. The model also produced realistic vertical distributions of conversion
rates, revealing clear depth-dependent zonation. Most marginal posterior distributions of estimated parameters were in good
agreement with their priors. Yet, strong deviations were observed for the N isotope effect associated with the first step of
denitrification, $\varepsilon_{Den1}$, which was estimated at ~2.8±1.1‰, significantly lower than the expected ~20‰. These findings were
confirmed by additional simulations performed using narrower priors and a fixed $\varepsilon_{Den1}$ value of 20‰, both of which resulted
in a substantial deterioration in the model's ability to reproduce $\delta^{15}$N-$NO_3^-$ profiles. This, in turn, can be taken as indication
for a suppressed denitrification $NO_3^-$ isotope effect at the porewater level in Lake Lucerne, potentially due to process
coupling via $NO_2^-$. The model's ability to quantify such interactions, which can be difficult to discern in situ or from field
data alone, is a key strength of this stepwise model framework.
Further sensitivity tests highlighted that the model could still achieve good fits to the observational data even when certain
individual processes were excluded, demonstrating the critical role of prior knowledge regarding estimated parameters and
their associated uncertainties.
Overall, this study presents the first comprehensive diagenetic N isotope model that explicitly incorporates multiple N
transformation pathways in a stepwise manner, validated against field measurements. Rather than serving as a purely
predictive tool, this model is intended to stimulate scientific discussion on the quantification of N transformations and
isotope dynamics in sediments based on observed data. Future developments could focus on improving identifiability
through additional, targeted observations, expanding model validation across distinct benthic environments, and the
incorporating additional isotope tracers, such as $\delta^{18}$O of $NO_3^-$ and $NO_2^-$, to further strengthen the model structure and
improve its reliability.





Appendix A: *Model processes and stoichiometry*

Table A1. **Overview of all modelled N-transformation pathways, including their stoichiometry and governing equations.** $R$ denotes the $^{15}N/^{14}N$ ratio derived from OM.

| Process | Step | $NH_4^+$ $^{14}N$ | $NH_4^+$ $^{15}N$ | $NO_2^-$ $^{14}N$ | $NO_2^-$ $^{15}N$ | $NO_3^-$ $^{14}N$ | $NO_3^-$ $^{15}N$ | $N_2O$ $^{14}N^{14}N$ | $N_2O$ $^{14}N^{15}N$ | $N_2O$ $^{15}N^{15}N$ | $N_2$ $^{14}N^{14}N$ | $N_2$ $^{14}N^{15}N$ | $N_2$ $^{15}N^{15}N$ | $O_2$ | $SO_4^{2-}$ | Rate |
|---|---|---|---|---|---|---|---|---|---|---|---|---|---|---|---|---|
| Oxic min. | | $\gamma_{MinOx}(1-R)$ | $\gamma_{MinOx}\,R$ | | | | | | | | | | | -1 | | $r'_{MinOx}\,[^{14}NO_3^-]$ |
| Denitrification | [1] | $\gamma_{Den1}(1-R)$ | $\gamma_{Den1}\,R$ | 1 | | -1 | | | | | | | | | | $r'_{Den1}\,[^{14}NO_3^-]$ |
| | | $\gamma_{Den1}(1-R)$ | $\gamma_{Den1}\,R$ | | 1 | | -1 | | | | | | | | | $r'_{Den1}\,[^{15}NO_3^-]\,(1-\varepsilon_{Den1})$ |
| | [2] | $2\gamma_{Den2}(1-R)$ | $2\gamma_{Den2}\,R$ | -2 | | | | 1 | | | | | | | | $r'_{Den2}\,[^{14}NO_2^-]\,[^{14}NO_2^-]$ |
| | | $2\gamma_{Den2}(1-R)$ | $2\gamma_{Den2}\,R$ | -1 | -1 | | | | 1 | | | | | | | $2\,r'_{Den2}\,[^{14}NO_2^-]\,[^{15}NO_2^-]\,(1-\varepsilon_{Den2})$ |
| | | $2\gamma_{Den2}(1-R)$ | $2\gamma_{Den2}\,R$ | | -2 | | | | | 1 | | | | | | $r'_{Den2}\,[^{15}NO_2^-]\,[^{15}NO_2^-]\,(1-\varepsilon_{Den2})^2$ |
| | [3] | $\gamma_{Den3}(1-R)$ | $\gamma_{Den3}\,R$ | | | | | -1 | | | 1 | | | | | $r'_{Den3}\,[^{14}14N_2O]$ |
| | | $\gamma_{Den3}(1-R)$ | $\gamma_{Den3}\,R$ | | | | | | -1 | | | 1 | | | | $r'_{Den3}\,[^{14}15N_2O]\,(1-\varepsilon_{Den3})$ |
| | | $\gamma_{Den3}(1-R)$ | $\gamma_{Den3}\,R$ | | | | | | | -1 | | | 1 | | | $r'_{Den3}\,[^{15}15N_2O]\,(1-\varepsilon_{Den3})$ |
| Sulfate reduction | | $\gamma_{MinSulfRed}(1-R)$ | $\gamma_{MinSulfRed}\,R$ | | | | | | | | | | | | -1 | $r'_{MinSulfRed}$ |
| Anaerobic min. | | $1-R$ | $R$ | | | | | | | | | | | | | $r'_{MinAnae}$ |
| Nitrification | [1] | -1 | | 1 | | | | | | | | | | -1.5 | | $r'_{Nit1a}\,[^{14}NH_4^+]$ |
| | | | -1 | | 1 | | | | | | | | | -1.5 | | $r'_{Nit1a}\,[^{15}NH_4^+]\,(1-\varepsilon_{Nit1,NO2})$ |
| | | -2 | | | | | | 1 | | | | | | -2 | | $r'_{Nit1b}\,[^{14}NH_4^+]\,[^{14}NH_4^+]$ |
| | | -1 | -1 | | | | | | 1 | | | | | -2 | | $2\,r'_{Nit1b}\,[^{14}NH_4^+]\,[^{15}NH_4^+]\,(1-\varepsilon_{Nit1,N2O})$ |
| | | | -2 | | | | | | | 1 | | | | -2 | | $r'_{Nit1b}\,[^{15}NH_4^+]\,[^{15}NH_4^+]\,(1-\varepsilon_{Nit1,N2O})^2$ |
| | [2] | | | -1 | | 1 | | | | | | | | 0.5 | | $r'_{Nit2}\,[^{14}NO_2^-]$ |
| | | | | | -1 | | 1 | | | | | | | 0.5 | | $r'_{Nit2}\,[^{15}NO_2^-]\,(1-\varepsilon_{Nit2})$ |
| Anammox | [m] | -1 | | -1 | | | | | | | 1 | | | | | $r'_{Anam}\,[^{14}NH_4^+]\,[^{14}NO_2^-]$ |
| | | -1 | | | -1 | | | | | | | 1 | | | | $r'_{Anam}\,[^{14}NH_4^+]\,[^{15}NO_2^-]\,(1-\varepsilon_{Anam,NO2})$ |
| | | | | -1 | | | | | | | | 1 | | | | $r'_{Anam}\,[^{15}NH_4^+]\,[^{14}NO_2^-]\,(1-\varepsilon_{Anam,NH4})$ |
| | | | | | -1 | | | | | | | | 1 | | | $r'_{Anam}\,[^{15}NH_4^+]\,[^{15}NO_2^-]\,(1-\varepsilon_{Anam,NO2})\,(1-\varepsilon_{Anam,NH4})$ |
| | [s] | | | -1 | | 1 | | | | | | | | | | $f_{side}\,r'_{Anam}\,[^{14}NH_4^+]\,[^{14}NO_2^-]$ |
| | | | | | -1 | 1 | | | | | | | | | | $f_{side}\,r'_{Anam}\,[^{14}NH_4^+]\,[^{15}NO_2^-]\,(1-\varepsilon_{Anam,NO2})\,(1-\varepsilon_{Anam,side})$ |
| | | | | -1 | | | 1 | | | | | | | | | $f_{side}\,r'_{Anam}\,[^{15}NH_4^+]\,[^{14}NO_2^-]\,(1-\varepsilon_{Anam,NH4})$ |
| | | | | | -1 | | 1 | | | | | | | | | $f_{side}\,r'_{Anam}\,[^{15}NH_4^+]\,[^{15}NO_2^-]\,(1-\varepsilon_{Anam,NO2})\,(1-\varepsilon_{Anam,NH4})\,(1-\varepsilon_{Anam,side})$ |
| DNRA | [1] | $\gamma_{DNRA1}(1-R)$ | $\gamma_{DNRA1}\,R$ | 1 | | -1 | | | | | | | | | | $r'_{DNRA1}\,[^{14}NO_3^-]$ |
| | | $\gamma_{DNRA1}(1-R)$ | $\gamma_{DNRA1}\,R$ | | 1 | | -1 | | | | | | | | | $r'_{DNRA1}\,[^{15}NO_3^-]\,(1-\varepsilon_{DNRA1})$ |
| | [2] | $1+\gamma_{DNRA2}(1-R)$ | $\gamma_{DNRA2}\,R$ | -1 | | | | | | | | | | | | $r'_{DNRA2}\,[^{14}NO_2^-]$ |
| | | $\gamma_{DNRA2}(1-R)$ | $1+\gamma_{DNRA2}\,R$ | | -1 | | | | | | | | | | | $r'_{DNRA2}\,[^{15}NO_2^-]\,(1-\varepsilon_{DNRA2})$ |





$$r_{\mathrm{MinOx}} = k_{\mathrm{MinOx}}\frac{[O_2]}{K_{\mathrm{O2,MinOx}}+[O_2]}$$

$$r_{\mathrm{MinAnae}} = k_{\mathrm{MinAnae}}\frac{K_{\mathrm{NO3,MinAnae}}}{K_{\mathrm{NO3,MinAnae}}+[^{14}NO_3^-]+[^{15}NO_3^-]}\frac{K_{\mathrm{O2,MinAnae}}}{K_{\mathrm{O2,MinAnae}}+[O_2]}$$

$$r_{\mathrm{MinSulfRed}} = k_{\mathrm{MinSulfRed}}\frac{[SO_4^{2-}]}{K_{\mathrm{SO4,MinSulfRed}}+[SO_4^{2-}]}\frac{K_{\mathrm{NO3,MinSulfRed}}}{K_{\mathrm{NO3,MinSulfRed}}+[NO_3^-]}\frac{K_{\mathrm{O2,MinSulfRed}}}{K_{\mathrm{O2,MinSulfRed}}+[O_2]}$$
$$r'_{\mathrm{Anam}} = k_{\mathrm{Anam}}\frac{1}{K_{\mathrm{NH4,Anam}}+[^{14}NH_4^+]+[^{15}NH_4^+]}\frac{1}{K_{\mathrm{NO2,Anam}}+[^{14}NO_2^-]+[^{15}NO_2^-]}\frac{K_{\mathrm{O2,Anam}}}{K_{\mathrm{O2,Anam}}+[O_2]}$$
$$r'_{\mathrm{Nit1a}} = k_{\mathrm{Nit1}}\left(1-f_{\mathrm{N2O,Nit1}}\right)\frac{1}{K_{\mathrm{NH4,Nit1}}+[^{14}NH_4^+]+[^{15}NH_4^+]}\frac{[O_2]}{K_{\mathrm{O2,Nit1}}+[O_2]}$$

$$r'_{\mathrm{Nit1b}} = k_{\mathrm{Nit1}}\, f_{\mathrm{N2O,Nit1}}\frac{1}{\left(K_{\mathrm{NH4,Nit1}}+[^{14}NH_4^+]+[^{15}NH_4^+]\right)^2}\frac{[O_2]}{K_{\mathrm{O2,Nit1}}+[O_2]}$$

$$r'_{\mathrm{Nit2}} = k_{\mathrm{Nit2}}\frac{1}{K_{\mathrm{NO2,Nit2}}+[^{14}NO_2^-]+[^{15}NO_2^-]}\frac{[O_2]}{K_{\mathrm{O2,Nit2}}+[O_2]}$$
$$r'_{\mathrm{Den1}} = k_{\mathrm{Den1}}\frac{1}{K_{\mathrm{NO3,Den1}}+[^{14}NO_3^-]+[^{15}NO_3^-]}\frac{K_{\mathrm{O2,Den1}}}{K_{\mathrm{O2,Den1}}+[O_2]}$$

$$r'_{\mathrm{Den2}} = k_{\mathrm{Den2}}\frac{1}{\left(K_{\mathrm{NO2,Den2}}+[^{14}NO_2^-]+[^{15}NO_2^-]\right)^2}\frac{K_{\mathrm{O2,Den2}}}{K_{\mathrm{O2,Den2}}+[O_2]}$$

$$r'_{\mathrm{Den3}} = k_{\mathrm{Den3}}\frac{1}{K_{\mathrm{N2O,Den3}}+[^{1414}N_2O]+[^{1515}N_2O]}\frac{K_{\mathrm{O2,Den3}}}{K_{\mathrm{O2,Den3}}+[O_2]}$$
$$r'_{\mathrm{DNRA1}} = k_{\mathrm{DNRA1}}\frac{1}{K_{\mathrm{NO3,DNRA1}}+[^{14}NO_3^-]+[^{15}NO_3^-]}\frac{K_{\mathrm{O2,DNRA1}}}{K_{\mathrm{O2,DNRA1}}+[O_2]}$$

$$r'_{\mathrm{DNRA2}} = k_{\mathrm{DNRA2}}\frac{1}{K_{\mathrm{NO2,DNRA2}}+[^{14}NO_2^-]+[^{15}NO_2^-]}\frac{K_{\mathrm{O2,DNRA2}}}{K_{\mathrm{O2,DNRA2}}+[O_2]}$$

$$f_{\mathrm{N2O,Nit1}} = b_{\mathrm{N2O,Nit1}}\frac{a_{\mathrm{N2O,Nit1}}}{a_{\mathrm{N2O,Nit1}}+[O_2]}$$
$$k_{\mathrm{Den2}} = f_{\mathrm{Den2,Den1}}k_{\mathrm{Den1}} \qquad k_{\mathrm{Den3}} = f_{\mathrm{Den3,Den1}}k_{\mathrm{Den1}} \qquad k_{\mathrm{Nit2}} = f_{\mathrm{Nit2,Nit1}}k_{\mathrm{Nit1}}$$
$$k_{\mathrm{Anam}} = f_{\mathrm{Anam,Den2}}k_{\mathrm{Den2}} \qquad k_{\mathrm{DNRA1}} = f_{\mathrm{DNRA1,Den1}}k_{\mathrm{Den1}} \qquad k_{\mathrm{DNRA2}} = f_{\mathrm{DNRA2,Den2}}k_{\mathrm{Den2}}$$



**Appendix B:** *Reaction-diffusion model*
Nomenclature
$t$        time [d]
$z$        depth coordinate within sediment (0 at the sediment surface, $d$ at the lower boundary of the modelled sediment
layer) [cm]

$d$        depth of the modelled sediment layer [cm]
$C(z,t)$    substance concentration (mass per volume of water) as a function of depth and time
$p(z)$    porosity of the sediment (water volume divided by sediment volume) as a function of sediment depth
$D(z)$    diffusivity of the substance in the water as a function of depth (usually constant and equal to the molecular
diffusion coefficient; however, bioturbation could be modelled as an increase in diffusivity close to the sediment
surface)

$r(C)$    transformation rate of the substance (mass per volume of water per unit of time)
$C_0$    substance concentration at the sediment surface
$F_d$    substance flux from deep sediment into the modelled sediment layer at the lower boundary of the modelled
sediment layer (mass per unit of total sediment surface and per unit of time)

Partial Differential Equation for Sediment Layer
Mass balance within the sediment layer:
$$p\frac{\partial C}{\partial t} - \frac{\partial}{\partial z}\left(D\,p\,\frac{\partial C}{\partial z}\right) = p\,r$$

Differential equation for concentration:
$$\frac{\partial C}{\partial t} = \frac{1}{p}\frac{\partial}{\partial z}\left(D\,p\,\frac{\partial C}{\partial z}\right) + r$$

Diffusion (molecular diffusion corrected for tortuosity, and bioturbation):
$$D = \frac{D_{\mathrm{mol}}}{a_{\mathrm{tort}}p^{1-m_{\mathrm{tort}}}} + D_{\mathrm{bio}}\,e^{-\frac{z}{d_{\mathrm{bio}}}}$$

Boundary conditions:
$$C(0,t) = C_0 \,, \ \ D(d,t)p(d,t)\frac{\partial C}{\partial z}(d,t) = F_d$$


For N compounds with a single N atom, the boundary conditions are calculated from total concentrations, $C_{tot}$, and $\delta^{15}N$ as
follows:
$$r = \left(\frac{\delta^{15}N}{1000} + 1\right)R_{std} \quad C_{^{14}N} = \frac{1}{1+r}C_{tot} \quad C_{^{15}N} = \frac{r}{1+r}C_{tot}$$

For N compounds with two N atoms, the boundary conditions are calculated from total concentrations, $C_{tot}$, and $\delta^{15}N$ as
follows (Drury et al., 1987):
$$r = \left(\frac{\delta^{15}N}{1000} + 1\right)R_{std} \ \ C_{^{14}N^{14}N} = \frac{1}{1 + 2r + r^2}C_{tot} \ \ C_{^{15}N^{14}N} = \frac{2r}{1 + 2r + r^2}C_{tot} \ \ C_{^{15}N^{15}N} = \frac{r^2}{1 + 2r + r^2}C_{tot}$$





**Appendix C:** *Prior values for inference*
**Table C1. Model parameters estimated using Bayesian inference, alongside their prior values and associated uncertainties.**
**Parameters are grouped into three categories: (A) reaction rates parameters (i.e., defining process kinetics), (B) isotope**
**parameters (i.e., isotope effects for the modelled processes and the N isotopic composition of OM), and (C) parameters used in the**
**one-step denitrification approach (NO₃⁻ → N₂ instead of NO₃⁻ → NO₂⁻ → N₂O → N₂). Where a wide range of values was reported**
**in the literature, the most relevant value for benthic environments was selected, and the corresponding reference is reported.**

| Description | | | Symbol | Distribution | Mean | St.deviation | Reference(s) |
|---|---|---|---|---|---|---|---|
| (A) *Reaction rate parameters* | | | | | | | |
| Aerobic mineralization | | Maximum conversion rate | $k_{MinOx}$ | Uniform | – | – | – |
| | | O₂ limitation constant | $K_{O2,MinOx}$ | Lognormal | 8 μM | 20% | (Rooze and Meile, 2016) |
| | | Fraction of NH₄⁺ produced | $\gamma_{NH4,MinOx}$ | Lognormal | 0.1509 | 10% | Stoichiometry |
| Anaerobic mineralization | | Maximum conversion rate | $k_{MinAnae}$ | Uniform | – | – | – |
| | | O₂ limitation constant | $K_{O2,MinAnae}$ | Lognormal | 5 μM | 20% | (Paraska et al., 2011) |
| | | NO₃⁻ limitation constant | $K_{NO3,MinAnae}$ | Lognormal | 5 μM | 20% | (Paraska et al., 2011) |
| Sulfate reduction coupled to mineralization | | Maximum conversion rate | $k_{MinSulfRed}$ | Uniform | – | – | – |
| | | O₂ limitation constant | $K_{O2,MinSulfRed}$ | Lognormal | 5 μM | 20% | Assumed to be comparable to $K_{O2,MinAnae}$ |
| | | NO₃⁻ limitation constant | $K_{NO3,MinSulfRed}$ | Lognormal | 5 μM | 20% | Assumed to be comparable to $K_{NO3,MinAnae}$ |
| | | SO₄²⁻ limitation constant | $K_{SO4,MinSulfRed}$ | Lognormal | 20 μM | 20% | (Richards and Pallud, 2016) |
| | | Fraction of NH₄⁺ produced | $\gamma_{NH4,MinSulfRed}$ | Lognormal | 0.3019 | 10% | Stoichiometry |
| Nitrification | [1] | Maximum conversion rate | $k_{Nit1}$ | Uniform | – | – | – |
| | | O₂ limitation constant | $K_{O2,Nit1}$ | Lognormal | 3.5 μM | 20% | (Martin et al., 2019) |
| | | NH₄⁺ limitation constant | $K_{NH4,Nit1}$ | Lognormal | 2.0 μM | 20% | (Wyffels et al., 2004) |
| | | N₂O production | $a$ | Lognormal | 0.2 μM | 10% | (Ji et al., 2018) |
| | | Maximum N₂O production | $b$ | Lognormal | 0.08 | 10% | (Ji et al., 2018) |
| | [2] | Reaction rate factor | $f_{Nit2}$ | Lognormal | 1 | 50% | |
| | | O₂ limitation constant | $K_{O2,Nit2}$ | Lognormal | 0.8 μM | 20% | (Martin et al., 2019) |
| | | NO₂⁻ limitation constant | $K_{NO2,Nit2}$ | Lognormal | 0.8 μM | 20% | (Wyffels et al., 2004) |
| Denitrification | [1] | Maximum conversion rate | $k_{Den1}$ | Uniform | – | – | – |
| | | O₂ inhibition constant | $K_{O2,Den1}$ | Lognormal | 3 μM | 20% | (Wenk et al. 2014) |
| | | NO₃⁻ limitation constant | $K_{NO3,Den1}$ | Lognormal | 2.46 μM | 20% | (Su et al., 2023) |
| | | Fraction of NH₄⁺ produced | $\gamma_{NH4,Den1}$ | Lognormal | 0.0755 | 10% | Stoichiometry |
| | [2] | Reaction rate factor | $f_{Den2}$ | Lognormal | 3 | 50% | |
| | | O₂ inhibition constant | $K_{O2,Den2}$ | Lognormal | 3 μM | 20% | Assumed to be comparable to $K_{O2,Den1}$ |
| | | NO₂⁻ limitation constant | $K_{NO2,Den2}$ | Lognormal | 0.41 μM | 20% | (Su et al., 2023) |
| | | Fraction of NH₄⁺ produced | $\gamma_{NH4,Den2}$ | Lognormal | 0.0755 | 10% | Stoichiometry |
| | [3] | Reaction rate factor | $f_{Den3}$ | Lognormal | 3 | 50% | |
| | | O₂ inhibition constant | $K_{O2,Den3}$ | Lognormal | 0.1 μM | 20% | (Suenaga et al., 2018) |
| | | N₂O limitation constant | $K_{N2O,Den3}$ | Lognormal | 3.7 μM | 20% | (Suenaga et al., 2018) |
| | | Fraction of NH₄⁺ produced | $\gamma_{NH4,Den3}$ | Lognormal | 0.0755 | 10% | Stoichiometry |
| DNRA | [1] | Reaction rate factor | $f_{DNRA1,Den1}$ | Lognormal | 0.005 | 25% | [15]N-tracer incubations (this study) |
| | | O₂ inhibition constant | $K_{O2,DNRA1}$ | Lognormal | 3 μM | 20% | Assumed to be comparable to $K_{O2,Den1}$ |



| | | | | | | | |
|---|---|---|---|---|---|---|---|
| | | NO$_3^-$ limitation constant | $K_{NO3,DNRA1}$ | Lognormal | 2.46 µM | 20% | Assumed to be comparable to $K_{NO3,Den1}$ |
| | | Fraction of NH$_4^+$ produced | $\gamma_{NH4,DNRA1}$ | Lognormal | 0.0755 | 10% | Stoichiometry |
| | [2] | Reaction rate factor | $f_{DNRA2,Den2}$ | Lognormal | 0.005 | 25% | [15]N-tracer incubations (this study) |
| | | O$_2$ inhibition constant | $K_{O2,DNRA2}$ | Lognormal | 3 µM | 20% | Assumed to be comparable to $K_{O2,Den2}$ |
| | | NO$_2^-$ limitation constant | $K_{NO2,DNRA2}$ | Lognormal | 0.41 µM | 20% | Assumed to be comparable to $K_{NO2,Den2}$ |
| | | Fraction of NH$_4^+$ produced | $\gamma_{NH4,DNRA2}$ | Lognormal | 0.226 | 10% | Stoichiometry |
| Anammox | | Reaction rate factor | $f_{Anam,Den2}$ | Lognormal | 0.2 | 25% | [15]N-tracer incubations (this study) |
| | | O$_2$ inhibition constant | $K_{O2,Ana}$ | Lognormal | 2.5 µM | 20% | (Kalvelage et al., 2011) |
| | | NH$_4^+$ limitation constant | $K_{NH4,Ana}$ | Lognormal | 1 µM | 20% | (Wenk et al. 2014) |
| | | NO$_2^-$ limitation constant | $K_{NO2,Ana}$ | Lognormal | 5 µM | 20% | Reported for NO$_3^-$ (Wenk et al. 2014) |
| | | NO$_3^-$ production factor | $f_{Anam, side}$ | Lognormal | 0.3 | 10% | (Brunner et al., 2013) |

| (B) *Isotope effects and $\delta^{15}N$* | | | | | | | |
|---|---|---|---|---|---|---|---|
| Nitrification | [1a] | NH$_4^+$ → NO$_2^-$ | $\varepsilon_{Nit1,NO2}$ | Normal | 30‰ | 5‰ | (Dale et al., 2022; Denk et al., 2017) |
| | [1b] | NH$_4^+$ → N$_2$O | $\varepsilon_{Nit1,N2O}$ | Normal | 40‰ | 5‰ | (Denk et al., 2017) |
| | [2] | NO$_2^-$ → NO$_3^-$ | $\varepsilon_{Nit2}$ | Normal | -13‰ | 5‰ | (Denk et al., 2017) |
| Denitrification | [1] | NO$_3^-$ → NO$_2^-$ | $\varepsilon_{Den1}$ | Normal | 20‰ | 5‰ | (Rooze and Meile 2016; A. W. Dale et al. 2019) |
| | [2] | NO$_2^-$ → N$_2$O | $\varepsilon_{Den2}$ | Normal | 15‰ | 5‰ | (Dale et al., 2019; Denk et al., 2017) |
| | [3] | N$_2$O → N$_2$ | $\varepsilon_{Den3}$ | Normal | 9‰ | 5‰ | (Wenk et al. 2016) |
| DNRA | [1] | NO$_3^-$ → NO$_2^-$ | $\varepsilon_{DNRA1}$ | Normal | 20‰ | 5‰ | (Rooze and Meile 2016; A. W. Dale et al. 2019) |
| | [2] | NO$_2^-$ → NH$_4^+$ | $\varepsilon_{DNRA2}$ | Normal | 15‰ | 5‰ | Assumed to be comparable to $\varepsilon_{Den2}$ |
| Anammox | | NH$_4^+$ → N$_2$ | $\varepsilon_{Anam,NH4}$ | Normal | 23‰ | 5‰ | (Brunner et al., 2013) |
| | | NO$_2^-$ → N$_2$ | $\varepsilon_{Anam,NO2}$ | Normal | 16‰ | 5‰ | (Brunner et al., 2013) |
| | | NO$_2^-$ → NO$_3^-$ | $\varepsilon_{Anam\_side}$ | Normal | -31‰ | 5‰ | (Brunner et al., 2013) |
| Organic Matter isotopic composition | | | $\delta^{15}N\text{-}OM$ | Normal | 3‰ | 0.5‰ | (Baumann et al., 2024) |

| (C) *One-step denitrification* | | | | | | | |
|---|---|---|---|---|---|---|---|
| Denitrification | | Maximum conversion rate | $k_{Den}$ | Uniform | – | – | – |
| | | O$_2$ inhibition constant | $K_{O2,Den}$ | Lognormal | 3 µM | 20% | (Wenk et al. 2014) |
| | | NO$_3^-$ limitation constant | $K_{NO3,Den}$ | Lognormal | 2.46 µM | 20% | (Su et al., 2023) |
| | | Fraction of NH$_4^+$ produced | $\gamma_{NH4,Den}$ | Lognormal | 0.189 | 10% | Stoichiometry |
| | | Isotope effect | $\varepsilon_{Den}$ | Normal | 20‰ | 5‰ | (Rooze and Meile 2016; A. W. Dale et al. 2019) |
| DNRA | [1] | Reaction rate factor | $f_{DNRA1,Den}$ | Lognormal | 0.005 | 25% | [15]N-tracer incubations (this study) |
| | [2] | Reaction rate factor | $f_{DNRA2,Den}$ | Lognormal | 0.005 | 25% | [15]N-tracer incubations (this study) |
| Anammox | | Reaction rate factor | $f_{Anam,Den}$ | Lognormal | 0.6 | 25% | [15]N-tracer incubations (this study) |

**Appendix D:** *Model discretization*
We discretize the partial differential equations outlined in Appendix B using the Method of Lines. This approach involves
explicit discretization in space, followed by the application of an ODE solver to the resulting system of ODEs.
Spatial discretization
Numerical discretization of sediment layer (*n* cells, cell expansion factor *f*):

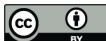



Visualization:

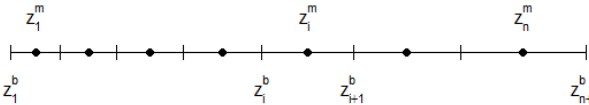


Cell boundaries ($i = 1, \ldots, n + 1$):
$$z_i^b = \begin{cases} \dfrac{i-1}{n}d & \text{for } f < 1.1 \quad (i = 1, \ldots, n + 1) \\[2mm] \dfrac{f^{\frac{i-1}{n}} - 1}{f - 1}d & \text{for } f \geq 1.1 \quad (i = 1, \ldots, n + 1) \end{cases}$$

Cell midpoints ($i = 1, \ldots, n$):
$$z_i^m = \frac{1}{2}\left(z_i^b + z_{i+1}^b\right)$$

Explanation for the cell expansion factor:
The cell size is approximately (the larger $n$ the closer) proportional to
$$\frac{\partial z_i^b}{\partial i} = \frac{\partial}{\partial i}\left(\frac{f^{\frac{i-1}{n}} - 1}{f - 1}d\right) = \frac{\log(f)}{f - 1}\frac{1}{n}f^{\frac{i-1}{n}}d$$

Comparing these cell sizes at the lower and upper boundaries leads to
$$\frac{\left.\dfrac{\partial z_i^b}{\partial i}\right|_{i=n+1}}{\left.\dfrac{\partial z_i^b}{\partial i}\right|_{i=1}} = f$$

This expression clarifies the meaning of the cell expansion factor (approximately equal to the ratio of cell size of lowest to
uppermost cell).
Discretized Ordinary Differential Equations
Mass balance within sediment layer cells ($i = 2, \ldots, n - 1$):
$$p(z_i^m)\frac{\partial C}{\partial t}(z_i^m)\left(z_{i+1}^b - z_i^b\right)$$

$$= -p(z_i^b)D(z_i^b)\frac{C(z_i^m) - C(z_{i-1}^m)}{z_i^m - z_{i-1}^m} + p(z_{i+1}^b)D(z_{i+1}^b)\frac{C(z_{i+1}^m) - C(z_i^m)}{z_{i+1}^m - z_i^m}$$

$$+ p(z_i^m)r(z_i^m)\left(z_{i+1}^b - z_i^b\right)$$

Differential equation for concentrations at cell midpoints of inner cells ($i = 2, \ldots, n - 1$):
$$\frac{\partial C}{\partial t}(z_i^m) = \frac{-p(z_i^b)D(z_i^b)\dfrac{C(z_i^m) - C(z_{i-1}^m)}{z_i^m - z_{i-1}^m} + p(z_{i+1}^b)D(z_{i+1}^b)\dfrac{C(z_{i+1}^m) - C(z_i^m)}{z_{i+1}^m - z_i^m}}{p(z_i^m)\left(z_{i+1}^b - z_i^b\right)} + r(z_i^m)$$

Boundary conditions:
$$C(z_1^b) = C_0, \quad D(z_{n+1}^b, t)p(z_{n+1}^b, t)\frac{C(z_{n+1}^b) - C(z_n^m)}{z_{n+1}^b - z_n^m} = F_d$$

$$\rightarrow \quad C(z_{n+1}^b) = C(z_n^m) + F_d\frac{z_{n+1}^b - z_n^m}{D(z_{n+1}^b, t)p(z_{n+1}^b, t)}$$

Differential equations for concentrations at cell midpoints of top and bottom cell ($i = 1, i = n$):



$$\frac{\partial C}{\partial t}(z_1^{\mathrm{m}}) = \frac{-p(z_1^{\mathrm{b}})D(z_1^{\mathrm{b}})\frac{C(z_1^{\mathrm{m}})-C(z_1^{\mathrm{b}})}{z_1^{\mathrm{m}}-z_1^{\mathrm{b}}} + p(z_2^{\mathrm{b}})D(z_2^{\mathrm{b}})\frac{C(z_2^{\mathrm{m}})-C(z_1^{\mathrm{m}})}{z_2^{\mathrm{m}}-z_1^{\mathrm{m}}}}{p(z_1^{\mathrm{m}})(z_2^{\mathrm{b}}-z_1^{\mathrm{b}})} + r(z_1^{\mathrm{m}})$$


$$\frac{\partial C}{\partial t}(z_n^{\mathrm{m}}) = \frac{-p(z_n^{\mathrm{b}})D(z_n^{\mathrm{b}})\frac{C(z_n^{\mathrm{m}})-C(z_{n-1}^{\mathrm{m}})}{z_n^{\mathrm{m}}-z_{n-1}^{\mathrm{m}}} + p(z_{n+1}^{\mathrm{b}})D(z_{n+1}^{\mathrm{b}})\frac{C(z_{n+1}^{\mathrm{m}})-C(z_n^{\mathrm{m}})}{z_{n+1}^{\mathrm{b}}-z_n^{\mathrm{m}}}}{p(z_n^{\mathrm{m}})(z_{n+1}^{\mathrm{b}}-z_n^{\mathrm{b}})} + r(z_n^{\mathrm{m}})$$


$$= \frac{-p(z_n^{\mathrm{b}})D(z_n^{\mathrm{b}})\frac{C(z_n^{\mathrm{m}})-C(z_{n-1}^{\mathrm{m}})}{z_n^{\mathrm{m}}-z_{n-1}^{\mathrm{m}}} + F_d}{p(z_n^{\mathrm{m}})(z_{n+1}^{\mathrm{b}}-z_n^{\mathrm{b}})} + r(z_n^{\mathrm{m}})$$


**Appendix E:** *Model implementation*
The model was implemented in Julia (Bezanson et al., 2017) (https://julialang.org). The implementation is available with
open access at https://gitlab.com/p.reichert/Nsediment. The version used for this study corresponds to commit
7afecdf1af871e8f8030360d658ec1cf54d20716.
The partial differential equations described in Appendix B were spatially discretized according to the approach outlined in
Appendix D. The resulting ordinary differential equations were then numerically solved by the Method of Lines using the
package DifferentialEquations.jl (Rackauckas and Nie, 2017). Discretizing the modelled sediment layer into 50 cells, and
considering 14 state variables, resulted in a system of 700 ordinary differential equations. The performance of several ODE
solvers was compared, resulting in the use of the adaptive order and adaptive time step backward-differencing solver FBDF
to account for the stiffness of the ODE system.
Maintaining compatibility with automatic differentiation while allowing flexible parameter selection for inference was a key
implementation challenge. This was addressed by using separate arrays for parameter values and names, and by prepending
the parameters to be estimated, ensuring a contiguous array of the parameters. To avoid inefficiencies related to the search of
parameter names, the association of parameter names to array indices was resolved within the differential equation solver
function. This solver, which includes the function to calculate the right-hand side of the differential equation as an internal
function, ensures that the index resolution has to be done only once and remains available for all calls of the integrator by the
solver. This approach enabled compatibility of our implementation with the automatic differentiation package ForwardDiff.jl
(Revels et al., 2016).
Bayesian inference was implemented with both an adaptive Metropolis sampler from the AdaptiveMCMC package (Vihola,
2020) and the Hamiltonian Monte Carlo algorithm from the AdvancedHMC.jl package (Xu et al., 2020).
All model outputs were written to text files and post-processed using R (https://www.r-project.org).



**Code and data availability**
The code for the isotope model presented in this manuscript is available at https://gitlab.com/p.reichert/Nsediment (commit
7afecdf1af871e8f8030360d658ec1cf54d20716).
Field data, model outputs and re-processing scripts are available through zenodo at
https://doi.org/10.5281/zenodo.14913873.
**Supplement link**
Supplementary material is provided alongside this manuscript.
**Author contribution**
The research was initiated and conceptually designed by AM, PR, and MFL. All co-authors contributed to the
conceptualization of the model, AM and PR developed the model code and performed the simulations. AM and PR prepared
the manuscript with input from all co-authors.
**Competing interests**
The authors declare that they have no conflict of interest.
**Acknowledgments**
Calculations were performed at sciCORE (http://scicore.unibas.ch/), the scientific computing centre at the University of
Basel. We thank Prof. Carsten Schubert for providing logistic support for access to Lake Lucerne, and the technical staff at
University of Basel and Eawag for their assistance with the field campaign and the resulting analytical work.
AI-based language tools were used on individual sentences to refine sentence structures and enhance the readability of the
manuscript.
**Financial support**
This study was funded by the Swiss National Science Foundation, grant SNSF 188728.

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
