# Peer review of "A comprehensive porewater isotope model for simulating benthic"

_EGUsphere, 2025_

## Author Comment (AC1)

Overall, I find this to be a worthy contribution to the field and suggest publication with minor changes. It does a good job of examining where measurement limitations are important in a complicated, under-determined system. I am not a modeling expert so I will not address the validity of their approach, other than to say that they do an impressive job of examining the multitude of process in sediment N cycling.

We thank Dr. Brandes for the kind and insightful feedback. We have assessed the implications of each comment and input, and will adapt the manuscript accordingly. Please refer to the points below for a detailed plan on how we intend to revise the text.

Specific comments- Line 61. Replace 'all' with 'present'. This century has seen tremendous advances in the ability to measure N cycle species, and it would be folly to state that these advances will not continue into the future.

We agree with Dr. Brandes' comment and will change "if not impossible at all" to "if not impossible at present" on Lines 60-61.

Fig 2 and other color plots. -These should be redone with an eye to increase legibility and distinction between parameters. It is quite difficult to distinguish between yellow, light orange and other similar colors, why not use a wider range?, and please consider those who are colorblind! The best practice is to assume that the reader might only have a greyscale printoff of the figure and make sure that your images are legible in grayscale.

We acknowledge the difficulty in identifying and distinguish the distinct profiles and will attempt to improve the graphics to the best of our possibilities. We have included here examples of Figure 2 (color palette: bright Paul Tol) and Figure 3 (color palette: muted Paul Tol) to show the potential improvements of graphs.

The concept of diffusion in sediments influencing the effective isotopic fractionation expressed in sediment (but not the intrinsic isotopic fractionation of denitrifiers) has been discussed widely in the literature, it is not at all a surprise that they find this as a requirement in their model. They may wish to better acknowledge this in their discussion/conclusions.

We appreciate Dr. Brandes' insightful comment emphasizing the well-established role of diffusion-limitation in influencing isotope dynamics within sediments. Indeed, we plan to submit a follow-up paper that provides an indepth assessment of isotope dynamics across several benthic habitats using the model, addressing this aspect in detail. We envision the current paper as a presentation of the model, focusing on its validation and technical aspects. While its main objective of the present paper was to introduce the model framework, we will nonetheless acknowledge the reviewer's input and explicitly address this point in the discussion/conclusions of the revised manuscript. Given the similarities with the comments from Reviewer #1, we will detail how we plan to edit the text in the RC1\_Reply file.

---

## Author Comment (AC2)

Mazzoli and colleagues present a model of benthic N cycling that includes a description of isotopic signatures. They apply it to a dataset obtained in Lake Lucerne, discuss the dominant processes in the nitrogen cycle, and make the case that the model is broadly applicable.

The combination or measured porewater profiles of N2, NH4 and NO3, together with their isotopic signatures and rates of denitrification, DNRA and anammox with the model is novel. What also sets this paper apart from previous work and many other early diagenetic modeling efforts is that it uses a strong parameter estimation component. They use Bayesian inference to connect prior knowledge with the data, and carry out sensitivity analyses on parameters for which the marginal posterior distributions differed substantially from the prior.

This is a well-written paper, providing a good overview on biogeochemical N models in the introduction. Overall, I really like this paper. The work seems to be technically sound, the analysis related to the model parameters is excellent (see e.g. the first paragraph of the conclusions). I also appreciate the clear demonstration that model fit to the measured profiles is not a strong indicator that the underlying processes are necessarily captured correctly (conclusion line 762ff).

We thank the reviewer for the kind and insightful feedback. We took the time to assess the implications of each comment. We plan to consider all of them and will adapt the manuscript accordingly. Please refer to the points below for a detailed plan on how we intend to revise the text.

The point that I struggled with most is the very low isotope effect reported for denitrification, which I am a bit skeptical about given this is model-based and lacks direct observational support. However, the model analysis presented in the paper is well thought out, so I wonder about potential assumptions that could lead to such a finding in the model (but potentially not in nature). One such issue might be that the abundance of active cells is not modeled (assumption iii, which is common). This may lead to large variations in cell specific rates and potentially fractionation effects. Are these likely candidates for the "structural limitations" mentioned on Line 398? If so, consider expanding on this in the discussion of your result. For example, rather low fractionation has been reported by Perez-Rodriguez et al. 2017 (https://doi.org/10.1016/j.gca.2017.05.014) and by Kritee et al. 2012 (https://doi.org/10.1016/j.gca.2012.05.020), and Kritee's figure 1 suggests lower epsilons at low cell specific rates. Nevertheless, the values reported here are much lower, so a broader discussion of existing supporting or contradictory experimental evidence would be welcome (including past work by some of the authors).

We are aware of the unexpectedly low isotope effects for NO3- reduction via denitrification, and have investigated this phenomenon extensively to ensure its robustness. We can assure the reviewer of the validity of our modeling

framework by presenting the case of a deep-sea station in the Bering Sea, originally reported in Lehmann et al. (2007; doi:10.1016/j.gca.2007.07.025), where we estimated  $^{15}\epsilon_{Den1}$  to be approximately 21‰ (i.e., the model is capable of capturing high porewater-level NO $_3$  reduction isotope effects when they occur). We speculate that the release of NO $_2$  into porewater during the first step of denitrification plays a key role in lowering the estimated  $^{15}\epsilon_{Den1}$  in Lake Lucerne sediments, as shown by the comparison of the one-step versus three-step denitrification approaches presented in section 4.4. A more extensive assessment of these isotope dynamics across a variety of benthic environments will be presented in a follow-up paper.

The main objective of the present paper is to introduce and validate the model framework, including technical details. Nonetheless, we value the suggested input on assumption (iii), and the consequent effects on cell-specific rates and non-constant isotope effects over depth, and will explicitly add that we did not consider variability in cell-specific rates under this assumption. While these considerations are valid, our measured  $\delta^{15}$ N-NO3 profile lacks a sharp gradient below the oxycline, suggesting a suppressed isotope effect for denitrification also in natural settings and not due to model artifacts. This is further confirmed by the comparison of model outputs for the one-step and three-step denitrification approaches (section 4.4). During this assessment, the model assumptions regarding cell densities and rates were kept unaltered, indicating that the primary reason for the low estimated  $^{15}\epsilon_{\rm Den1}$  in the three-step denitrification approach likely lies in the release of NO2 during denitrification.

Lastly, the "structural limitations" mentioned on Line 398 (in the original MS) refer to all model assumptions listed in section 2.3, including processes not explicitly considered within the model, and the parameterization of modelled processes. These structural limitations increase model uncertainty relative to observation error, which combines sampling and analysis uncertainties.

Below are a few more comments:

**Model formulation**

The mineralization of organic matter is modeled depending on intrinsic rate constants for aerobic and anaerobic mineralization reactions, as well as for sulfate reduction and denitrification. Different pathways are modeled depending on the availability of dissolved electron acceptors, as well as inhibition of anaerobic mineralization by O2 and NO3. Notably, the mineralization rate is modeled to not depend on the availability of reactive organic matter.

While this manuscript clearly expands beyond previous work, I suggest to revise the wording on line 77-79 and 765/766. For example, the work by Rooze and Meile also incorporated (some) stepwise process descriptions that emphasize the role of nitrite

as intermediate, and uses data (though not porewater profiles) for validation. Also, some existing models deal more comprehensively with the coupling between different elemental cycles (see below), and the applicability of this particular model to other environments really depends on how substantial this coupling is. For example, I did not see how reduced products from anaerobic mineralization reactions are accounted for (other than ammonium). As a consequence, the model will underestimate the use of nitrate and O2 through their reoxidation reaction. It seems that in the particular application for Lake Lucerne, this may not affect the results greatly (compare rates of anaerobic mineralization to those of aerobic mineralization in Figure 3, indicating that there is little anaerobic mineralization). However, these are modeled rates, and not accounting for the use of O2 to oxidize hydrogen sulfide or reduced metals limits the applicability of the model to a broader range of environments.

We thank the reviewer for the insightful input on the coupling of different element cycles and their role in the sedimentary redox zone. While we aimed to make the model as comprehensive as possible, the large number of parameters and limited field measurement data currently define the feasibility limits of the parameter estimation with adequate accuracy. As the model code is publicly available on GitLab, other users are encouraged to revise and extend the model for their specific needs, including the addition of new state variables and reactions, provided that they can be constrained by field data.

Taking these considerations into account, we will update the manuscript accordingly (line numbers still refer to the original manuscript):

- L77-79: "To date, only limited efforts have been made to develop comprehensive benthic isotope models that integrate multiple N-transformation processes in a stepwise manner, and assess the expression of their isotope effects in the porewater of aquatic sediments, validated with observational data (Denk et al., 2017; Rooze and Meile, 2016)."
- L765-767: "Overall, this study presents one of the first comprehensive diagenetic N isotope models that explicitly incorporate multiple transformation pathways in a stepwise manner and are validated against field measurements."
- Add a point to the list of model assumptions in section 2.3 stating that we do not consider re-oxygenation of reduced compounds other than  $NH_4^+$ .

All state variables considered I believe are solutes, i.e. the code does not track any solids. This shortens the simulation time and makes the MCMC feasible. However, it also implies that bioturbation does not account for solid phase mixing. Thus, it is

not surprising that changing Db has largely no effect on the model results (section 4.2, scenario D). As a consequence, I suspect this is largely a result caused by the model structure and unlikely to be true in reality. Furthermore, it is not clear how the value of Dbio was determined. Its current value drops from 1.16e-9 m2/s at z=0 to 4.26e-10 m2/s at 1cm depth. Thus, it essentially only exceeds molecular diffusion in the top cm. However, non-diffusive mixing typically exceeds solute transport by molecular diffusion more substantially, and bioturbation tends to often dominate the movement of solid phases near the sediment-water interface. Thus, this value should be justified for solutes and if this is hard to do, then consider toning down conclusions related to bioturbation. And because there are no solid phases accounted for, the model also ignores the effect of the precipitation of FeS (maybe not a huge pool, but reported present (e.g. Horw Bay; Table 2 in Urban et al. 1999 https://doi.org/10.1016/S0016-7037(98)00306-8), affecting the extent of use of electron acceptors such as nitrate and dissolved oxygen.

We agree with the reviewer that the complexity of modeling bioirrigation and its effects on solute diffusion extends beyond the simple diffusion enhancement represented in our model. Among other simplifications, our model does not take into account the solid phase, and we will emphasize this limitation in the model assumptions of the revised manuscript.

Our assessment of bioirrigation focused primarily on identifying the model sensitivity to bioturbation, as the magnitude of its effect on solute diffusion remains highly uncertain (as stated in section 4.2). However, we respectfully disagree with the reviewer's statement that "changing Db has largely no effect on the model results (section 4.2, scenario D)". While the effects on isotopic composition and fit quality was indeed small (Lines 532-533 of the original manuscript), altering the Db value (i.e., the magnitude of solute-diffusion enhancement due to bioirrigation) has a pronounced effect on the modelled rate parameters (Lines 530-531).

The manuscript says that rates of N cycling were measured, but I didn't see those results. The rate of (total) organic matter mineralization is on the order of 450 uM N/d near the surface, which translate into approximately 3 umol/L/d or 120 nmol/cm3/hr. (Figure 3: I assume these rates are in uM nitrogen/d because different N processes are being compared. Please clarify if these are in mol C). These rate estimates are about 100 times larger than the 1 nmolC/cm3/hr reported in Fiscal et al. 2019 (https://doi.org/10.5194/bg-16-3725-2019) for Lake Lucerne surface sediment. Please clarify the evidence supporting your numbers (measured rates; or fitting profiles using low estimates of transport as discussed above?)

We thank the reviewer for the careful attention to this comparison. We confirm that all rates are expressed in units of N and not C; therefore, the rates in Figure 3 are indeed uM N d-1. The conversion provided by the reviewer

appears to be off by a factor of  $10^3$  (i.e.,  $3 \mu mol/L/d = 0.125 nmol/cm3/h$ ), which brings our estimates much closer (within less than an order of magnitude) to those reported by Fiskal et al. (2019). Nonetheless, a key difference between our study and the study by Fiskal et al (2019) lies in the depth resolution of the profiles. As the oxycline and nitracline in the sediments of Lake Lucerne are located within the top 1-1.5 cm, our finer resolution allows us to better resolve these steep gradients and thereby better constrain N-cycling reactions occurring within the upper few mm of the sediment.

The measured rates were obtained using 15N tracer incubations, which provide rate estimates under above-natural substrate concentrations (potential rates). Thus, we used these data only to quantify the relative importance of denitrification, anammox and DNRA. The ratios among these processes were used as priors for the respective f factors listed in Table C1.

The model description is clear, but I have some questions about the process descriptions

- denitrification and nitrification: these processes are implemented as multi-step processes, with the rates of the steps following the initial step modeled as a fraction of the rate of that first step. This is done "to prevent unrealistic rates" (e.g. L180). But it also predetermines the "sharing" of NO2- as a key intermediate between different processes, and, for example, with the Monod dependency on nitrite, the second step of denitrification could not occur at a rate higher than what is fueled by nitrate. This may not be a huge issue for N dynamics and I consider this to be a reasonable model simplification, but can you discuss the implications in particular for the isotopic signatures?

This comment is based on a misunderstanding of our model parameterization, which we clarify below for denitrification.

Parameterizing the denitrification model with rate parameters  $k_{Den1}$ ,  $k_{Den2}$  and  $k_{Den3}$  is mathematically equivalent to our parameterization with  $k_{Den1}$ ,  $f_{Den2Den1}$  and  $f_{Den3Den1}$ , as these parameters are linked through the invertible transformations  $k_{Den2} = f_{Den2Den1}$   $k_{Den1}$  and  $k_{Den3} = f_{Den3Den1}$   $k_{Den1}$  (line 783). Therefore, identical results can be produced with both parameterizations for any values of  $k_{Den1}$ ,  $k_{Den2}$  and  $k_{Den3}$ . The reviewer seems to interpret the factors f as "fractions" smaller than 1, which is not the case; the priors (see Appendix C) do not constrain these values to be <1.

The issue of "unrealistic rates" arises because, during inference, the rate parameters of the second and third steps can grow indefinitely without producing unreasonable concentration profiles, as their rates are limited by the first reaction step. Increasing the rate parameters of the succeeding steps

thus only reduces the concentrations of intermediate reaction compounds without strongly modifying overall rates.

In Bayesian inference, such identifiability problems are addressed through prior information, since the data alone may not sufficiently constrain all rate parameters (unless the intermediate compound is measured and inconsistent with very small concentrations). The kDen rate coefficients can vary very strongly from one system to the other, as they are proportional to the product of the bacterial abundance and their growth rate, both of which were not explicitly modeled, are system specific and can vary widely. To account for this, we used a uniform prior for  $k_{Den1}$ . This approach, however, is not suitable for  $k_{Den2}$  and  $k_{Den3}$  for the reason mentioned above, so informative priors are needed. Because higher bacterial densities for the first step are typically correlated with higher densities in the following steps, the ratios kDen2/kDen1 (=  $f_{Den2Den1}$ ) and  $k_{Den3}/k_{Den1}$  (=  $f_{Den3Den1}$ ) are less variable between systems than the absolute rate coefficients ( $k_{Den2}$  and  $k_{Den3}$ ) themselves. Thus, it is easier to formulate "universal" priors for these ratios (the factors f) than for the absolute rates, preventing the inference process to "move" towards unrealistically large values without requiring assumptions about bacterial densities.

We will emphasize this reasoning in the revised manuscript by adding the following text: "The re-parameterization of the second and third steps using the  $f_{Den2Den1}$  and  $f_{Den3Den1}$  factors corresponds to exactly the same model without any approximation or simplification. It serves solely to facilitate the specification of priors, as more knowledge is typically available about ratios of maximum rates (i.e.,  $f_{Den2Den1} = k_{Den2}/k_{Den1}$ ) than about the absolute maximum rates themselves."

- the rate expression of reaction 1b (Table 1) follows a Monod dependency on both of the ammonia involved forming the N2O. This contrasts with pretty much all the other reactions (including the O2 dependency in the same reaction), in which the reaction stoichiometry is not accounted for in the rate laws. Is there any experimental evidence that supports this formulation? And maybe more importantly what are the implications of this choice?

We include limitations in all consumed compounds to prevent a process from continuing when one of the reactants reaches zero concentrations (as this would lead to negative concentrations calculated by the model). For reaction 1b, the quadratic dependence provides a meaningful limitation for *both* ammonia molecules at low concentrations.

While this is a convenient feature for the bulk model (which does not distinguish isotopes), it becomes absolutely essential if we distinguish

isotopes. Table 1 summarizes only the partial model for the  $^{14}$ N species. The complete model formulation in Table A1 expands to the combinations  $^{14}$ NH4+  $^{14}$ NH4+  $^{14}$ NH4+  $^{15}$ NH4+  $^{15}$ NH4+  $^{15}$ NH4+  $^{14}$ NH4+ and  $^{15}$ NH4+  $^{15}$ NH4+  $^{15}$ NH4+  $^{15}$ NH4+  $^{14}$ NH4+  $^{14}$ NH4+  $^{15}$ NH4+  $^{15}$ NH4+  $^{15}$ NH4+  $^{15}$ NH4+  $^{15}$ NH4+  $^{16}$ NH4+  $^$

The same argument applies to the second step of denitrification and is implemented analogously (see Table A1). We also note a misprint in Table 1, equation [2] for denitrification: [14NO2-] should read [14NO2-]2. We will correct this error.

As explained above, separate limitations are already needed for conceptual reasons (to formulate a consistent model); there is not much experimental evidence for the exact functional form of these limitation terms. This is also the case for all other limitation terms in the model. We also implemented exponential instead of Monod-type limitations in the model, but the results did not substantially change.

- anammox: this process includes not only the NO2 + NH4 —> N2 reaction, but also the production of nitrate from nitrite. If I understand correctly, the parameter fside (Table A1) is therefore representing the 0.3 NO3/NH4 (Line 205). Please clarify; also define [s] and [m] in table A1.

We confirm that the production of  $NO_3^-$  from  $NO_2^-$  is what we define with the parameter " $f_{\text{side}}$ ". We will revise the caption of Table A1 to clarify the two anammox reactions: "Anammox encompasses both the comproportionation of  $NH_4^+$  and  $NO_2^-$  to  $N_2$ , defined as the main ("m") reaction, and the production of  $NO_3^-$  from  $NO_2^-$ , defined as the side ("s") reaction".

To ensure clarity also in the text, we will include this information in section 2.2, where we describe the modelled transformation processes (Line 204): "Anammox is modelled to include both the comproportionation of  $NH_4^+$  and  $NO_2^-$  to  $N_2$  (main reaction, "m"), and the  $NO_3^-$  production via  $NO_2^-$  oxidation (side reaction, "s") (where 0.3 mol  $NO_3^-$  are produced per 1 mol  $NH_4^+$  and 1.3 mol  $NO_2^-$ )."

- Table A1: define gamma\_Den1, \_Den2, \_Den3. Please clarify so it is clear why R is 15N/14N, instead of e.g. 15N/(14N+15N).

We thank the reviewer for pointing out this this oversight. We will:

- clarify this by adding " $\gamma$ " on line 238 so that assumption (vii) will read: "OM composition is approximated by the Redfield ratio (C:N:P = 106:16:1), used to estimate the fraction of NH4+ released during OM mineralization,  $\gamma$ "
- revise the caption of Table A1 to include the meaning of  $\gamma$ : "The  $\gamma$  parameter defines the fraction of NH4+ released during OM mineralization for each reaction."
- change the caption to reflect the real definition of the R term as  $^{15}N/(^{14}N+^{15}N)$ .

Model/parameter analysis

The Bayesian inference analysis is well done and very helpful. For example, I was rather skeptical that a, b from the Ji et al. paper can be directly used in the sediment (p.8, 18). However, these parameters are apparently not impacting the results a great deal. This is a great example of the value of assessing the impact of the parameters on the model outcome.

We are grateful that the reviewer appreciates our efforts that went into validating and assessing the impact of the parameters on the model output. We emphasize that in other systems with different properties, the impact of each parameter may vary, and should be assessed prior to applying the model to isotope dynamics.

Visualization/presentation of results

Several figures are difficult to read without magnifying them on the screen. For example, in Figure 3 I had a difficult time identifying which line and process belong together. If another color scheme is not feasible, consider putting information identifying the relevant processes into the figure caption.

We acknowledge the difficulty in identifying and distinguishing the distinct profiles and will improve the graphics to the best of our ability. We will adopt a color-vision-deficiency-friendly palette, increase line widths and font size. We provide here an example of Figure 3 to show how we plan to change the figures to make it easier to distinguish the profiles.

Figure S1: spell out what the solid (posterior) and dashed (prior) lines represent, and what ess= ... means in the titles of each panel

We thank the reviewer for pointing out the missing information. We will revise the caption to "Figure S1. Marginal prior distributions (dashed) and marginal posterior distributions (solid) for all parameters estimated in the Base scenario. The effective sample size (ess; the approximate number of independent posterior sample points) for each parameter is also reported."

Line 242: in the reactions of the manuscript you refer to Mn2+, but here is it Mn3+.

We thank the reviewer for noting this inconsistency. We will replace "than oxidation by iron(III), Fe3+, and manganese, Mn3+, in some lacustrine systems" with "than oxidation by iron(III), Fe3+, and manganese, Mn4+, in some lacustrine systems" in lines 241-242. This is to ensure consistency with the equation provided in Table 1, where Mn4+ (in MnO2) is reduced to Mn2+. For clarity, our model does not include Mn(III) intermediates.

---

## Author Comment (AC3)

The authors present a diagenetic N-isotope model, for use in aquatic sediments. The model is fitted to data and used to estimate the magnitudes of various sedimentary processes. The manuscript is well-written, the model is novel, and the sensitivity analysis and model fitting is state of the art.

We appreciate the insightful feedback from the reviewer and have assessed the implications of each comment. Please refer to the points below for a detailed plan on how we intend to revise the text.

In contrast to other diagenetic models, this model considers only dissolved nitrogen species, imposing the mineralisation rates not by modelling organic matter, but rather by imposing the maximum rates of the separate processes. While the mineralisation processes comprise oxic mineralisation, denitrification and anoxic processes, the sulphate reduction is modelled separately. It is not clear why the authors have distinguished sulphate reduction from the anaerobic mineralisation (but I guess this is because sulphate was measured). However, one could think of a simpler model where sulphate reduction would be part of the anaerobic mineralisation.

We agree that separating sulfate reduction from anaerobic mineralization might not always be a sensible choice. First, in some lacustrine systems the rate of sulfate reduction can exceed rates of anaerobic mineralization by iron or manganese (Lines 241-242) (Steinsberger et al. 2020). Second, sulfate reduction commonly spans a thick layer of sediments in marine systems (Lines 242-243). Third, as mentioned by the reviewer, the availability of sulfate concentration data allowed separate identification sulfate reduction rates (i.e., separately from the other anaerobic mineralization processes). Based on these considerations, we deemed it reasonable to separate sulfate reduction from other anaerobic mineralization pathways in this application. Nonetheless, the model is modular: if future users prefer to have a lumped formulation for anaerobic mineralization, the rate of sulfate reduction,  $k_{\text{MinSulfRed}}$ , can easily be set to zero in the model, effectively merging it into the broader anaerobic mineralization term.

Ignoring organic matter in the model assumes that the mineralisation is only dependent on the availability of oxidants and not on organic matter. The anaerobic mineralisation is the closure term here and it is not limited by any substrate: it has only inhibition components (p.9). Hence, below the layers where oxygen and nitrate are present, anoxic mineralisation will continue at the same rate for all depths, and integrated anaerobic mineralisation will be infinitely large (theoretically). In the model, this is overcome by imposing an ammonium flux at the lower boundary, which effectively represents a \*finite\* ammonium production by anaerobic mineralisation. This means that the depth of the model is also an important model parameter, and so one cannot simply extend the model domain, and obtain the

same results, as one could do in other diagenetic models. This should be mentioned in the model assumptions section.

We agree that our assumption of sufficient readily degradable organic matter applies only within a reasonable depth range. At very large, effectively infinite depths, this assumption would no longer hold because mineralization (including anaerobic mineralization) would become substrate-limited. We already note the absence of explicit organic matter as a state variable and limiting factor in section 2.3 (assumptions i and ii). We will add an additional sentence to section 2.3 to clarify that this assumption is intended for layers with sufficient readily degradable organic matter (e.g., top 5 cm of the sediments in Lake Lucerne), and may break down at greater depths.

However, we respectfully disagree that the need for an  $NH_4^+$  flux at the lower boundary arises from not modelling organic matter. The observed  $NH_4^+$  profiles show a clear gradient at 5 cm depth that indicates an upward flux of  $NH_4^+$  in this depth, and for any model of this sediment layer, even one that explicitly accounts for organic matter, a lower boundary condition with an upward flux of  $NH_4^+$  is required. This flux as a lower boundary is not a problematic parameter of the model, as it is well informed by the data (and therefore identifiable even under a uniform prior), because the measured gradient constrains it directly.

We will clarify on line 159 that we estimated the ammonium flux at the lower boundary because the field data display a clear gradient, in contrast to the other state variables. We will also point out that the ammonium flux was estimated alongside  $\delta^{15}N_{FNH4}$ .

More seriously, is that, when looking at the description of oxygen dynamics, the reoxidation of anoxic substances other than ammonium and nitrite is ignored. This implicitly assumes that the concentrations of Fe2+, Mn2+, H2S, CH4 are completely removed in the deep parts of the sediment and therefore do not react with oxygen. While such removal processes may occur in certain sediments, it is rare that they completely remove all of these substances. The authors should also list this important (and perhaps unrealistic) assumption in the model assumptions section (on p 10). A suggestion to make the oxygen dynamics more robust would be to explicitly model a lump-sum of anoxic concentrations in the model that are reoxidised with oxygen, and impose a flux of this lump-sum constituent at the lower boundary.

We agree that it would be a conceptual improvement to include a lumped pool of reduced species and their partial re-oxidation in the upper sediments. However, we do not assume that these substances (Fe2+, Mn2+, H2S, CH4) are completely removed at depth (we agree that this would be unrealistic), as

shown by the benthic fluxes presented in Steinsberger et al. (2020) for Lake Lucerne. Nevertheless, their contribution to O2 consumption by re-oxidation processes in the top sediment layers is expected to be small relative to oxic mineralization and nitrification. We will add this point explicitly to the model assumptions.

"Re-oxidation of reduced species other than NH4+ and NO2- (e.g., Fe2+, Mn2+, H2S, CH4) is neglected in the oxygen budget for the modeled interval; this is appropriate where their upward fluxes are minor, but may underestimate O2 demand in settings with substantial reduced-species fluxes. Future users are encouraged to adapt the model to their research questions and dataset, including adding processes and state variables, provided that they can be constrained."

We note that adding these processes in the model would introduce additional poorly-identifiable model components without independent constraints, increasing uncertainty or requiring prior knowledge that we do not currently have. We have included a statement in the assumption above about the possibility of modifying the model for specific needs, including considering reoxidation of reduced species.

**Some minor comments:**

The figures were rather difficult to interpret, due to a color scheme that did not provide enough discriminating power. This made it difficult to follow the discussion.

See comment to Reviewer 1. We provide here an example of revised Figure 3 to show how we plan to change the color scheme to improve readability, while ensuring they remain color-vision-deficiency-friendly:

The ammonium deep boundary flux is imposed. How is this flux divided into 14N-NH4 and 15N-NH4?

We parameterized the influx of  $^{14}$ N-NH4+ and  $^{15}$ N-NH4+ using the total NH4+ flux (FNH4) and its  $\delta^{15}$ NFNH4. Both of these parameters are well identifiable from uniform priors, primarily because of the available profile data for NH4+ concentrations and  $\delta_{15N,NH4}$  (see Figure S1 for their marginal posteriors). We will add the prior values for FNH4 and  $\delta^{15}$ NFNH4 to Table C1 and will mention both of them being estimated on lines 158-159.

The model was dynamically run to steady-state. How was steady-state checked?

We simulated 100 days and plotted profiles every 10 days, which demonstrated that the steady-state was reached to an excellent approximation well before 100 days. Due to the adaptive time-stepping of our numerical integration algorithm, we could be generous with the choice of 100 days as this did not have an essential impact on simulation time because the time step becomes large when changes in the state variables become small.

L 99 sediment reactivity -> organic matter reactivity

We will change the text as suggested by the reviewer.

L176: manganese, iron -> manganese and iron oxides

We will change the text as suggested by the reviewer.

L775: [NO3-] instead of 14NO3- + 15NO3-?

We cannot find any mentioning of " $^{14}NO_3$ " +  $^{15}NO_3$ " on Line 775.

L330 Wagenigen -> Wageningen

We will change the text as suggested by the reviewer.

L508. A bioturbation coefficient of 1cm2/day seems to be very high.

Questions about the choice of the bioturbation coefficient were raised also by Reviewer 1 (so please also see our reply to R1). While we have information on the depth affected by bioturbation and an estimate of the bioturbator abundance per unit of volume (Fiskal et al. 2021), substantial uncertainty remains regarding the enhancement of solute diffusion due to their presence/activity. Our analysis is intended to assess model sensitivity to changing bioturbation; therefore, the exact value was not taken from literature but chosen as a representative case. Specifically, a bioturbation coefficient of 1 cm d-1 implies an effective solute diffusivity approximately

twice the molecular diffusivity. Nonetheless, we will add this clarification to the description of the "Enhanced bioturbation" scenario (Lines 508-512).

Table 1. The reaction for anammox produces organic matter; however this is not so for the nitrification reactions, which are also autotrophic.

The reviewer raises an important point: organic matter (biomass) production during bacterial growth is a key component of the benthic nitrogen cycling. In our framework, we do not consider/model bacterial growth explicitly (assumption iii); therefore, we formulated all processes (process rate laws) without biomass as state variable or limiting substrate. The only exception is anammox, because the NO2- oxidation to NO3- requires OM production to close the redox balance (Brunner et al. 2013).

In our model, as mentioned, OM is neither a state variable nor a limiting substrate, and we do not track OM production and consumption. Consequently, this point, while valid, does not affect the present model results. We will clarify this assumption further in the revised manuscript.

Table 1a: (1-ksi) -> (1-ksi/1000)

We respectfully disagree with the reviewer's comment. Our  $\varepsilon$  values (see Table C1) are 5‰ = 0.005, 20‰ = 0.020, etc. For this reason, another division by 1000 is not needed. The unit of "permille" includes the division by 1000 already.

Table C1 shows fractions of NH4 produced based on aerobic mineralisation, denitrification, DNRA and sulphate reduction. This does not seem to be consistent with the text where it is said that this is determined by the organic matter 15N/14N composition. Why not use the stoichiometry of the reactions to estimate the NH4 production from the OM composition?

We appreciate the opportunity to clarify this aspect. Indeed, in our model, the amount of  $NH_4^+$  produced ( $\gamma$ ) from OM composition is determined stoichiometrically from the OM composition and the reaction stoichiometries of the respective mineralization pathways. The respective  $\gamma$  values are reported in Table C1. On the other hand, the isotopic composition of the released  $NH_4^+$  is determined by the organic matter  $^{15}N/(^{14}N+^{15}N)$  composition. Table A1 shows the stoichiometry of all processes and clarifies where R ( $^{15}N/(^{14}N+^{15}N)$ ) composition of organic matter) and where the  $\epsilon$  values for isotope fractionation are used.

fNit2 present in table C2 is not in Table A1.

The  $f_{Nit2}$  term can be found in the equations below Table A1, specifically on Line 783 of the original manuscript.

The default parameter values for most parameters can be found in table C1, but not the rates, and the boundary conditions. All parameter values used for the base run should be presented somewhere in the manuscript.

We agree and will provide information on the boundary conditions (including the  $F_{NH4}$  and  $\delta^{15}N_{FNH4}$  mentioned in an earlier comment, which were obtained from field data) with units. As the system-specific rate constants (k values) are estimated using a uniform prior distribution, we will maintain the "-" in Table C1 for these parameters.

At the discretion of the editor, we will add a column to Table C1 with the estimated values (i.e., posterior distributions).